

# Analysis of instability conditions and failure mode of a special type of translational landslide using a long-period monitoring data: a case study of the Wobaoshi landslide(Bazhong city,China)

Yimin Liu [a,b] , Chenghu Wang [a,*] , Guiyun Gao [a] , Pu Wang [a] , Zhengyang Hou [a] , Qisong Jiao [a]

[a] Institute of Crustal Dynamics, China Earthquake Administration, Beijing, *100085*,China

[b] School of Manufacturing Science & Engineering, Sichuan University, *Chengdu,* 611730*,China*

**Abstract:** A translational landslide comprising nearly horizontal sand and mud interbed was widely developed in the Ba river basin of the Qinba–Longnan mountain area. Scholars have conducted theoretical research on this rainfall-induced landslide; however, owing to the lack of landslide monitoring engineering and data, demonstrating and validating the theoretical research wasdifficult. This study considered a translational landslide with an unusual morphology: the Wobaoshi landslide, which is located in Bazhong city, China. First, the formation conditions of this landslide were ascertained through field exploration, and the deformation and failure characteristics of the plate-shaped sliding body were analyzed. Then, long-period monitoring engineering was conducted to obtain multi-parameter monitoring data, such as crack width, rainfall intensity, and pore-water pressure. Finally, through the mechanical model analysis of the multi-stage sliding bodies, the calculating formula of the maximum height of the multi-stage plate girders, hcr, was derived, and the long-period monitoring data were used to verify its accuracy. Combined with numerical simulation and calculations, the deformation and failure modes of the plate-shaped sliding bodies were analyzed and explored. In this paper, the multi-parameter monitoring data proved that the stability of the sliding body is affected greatly by the rainfall



intensity and pore-water pressure and the pore-water pressure in the crack is positive for the
beginning of the plate-shaped sliding bodies, and an optimization monitoring method for this type
of landslide was proposed. Therefore, this paper has theoretical and practical significance for the
intensive study of translational landslides in this area.
**Keywords:**    Translational landslide; A Long- period monitoring; Instability conditions; Failure
mode; Plate-shaped sliding body; Pore-water pressure.
# 0. Introduction
A special type of landslide occurs in the red beds of Qinba–Longnan mountainous area. This
landslide is mainly developed in the rock mass of the nearly horizontal sand and mudstone
interbed in the Ba river basin. This phenomenon has the following characteristics: the cover layer
is extremely thin, generally not more than 5 m; the sliding surface is close to horizontal; and the
rock layer inclination angle is generally only $3°–8°$. The sliding body of this landslide is typically a
thick layer of sandstone with good integrity, and the bottom is a weak layer consisting of
mudstone. During the rainy season, particularly when rainstorms occur, the sliding body is pushed
horizontally along the sliding surface. Some scholars call this phenomenon a flat-push landslide,
which is a typical rainfall-induced landslide (Zhang et al., 1994; Fausto G. et al., 2004; Xu et al.,

2010).

Research on the formation mechanism and deformation mode of a translational landslide is
divided mainly into two perspectives. The first is the translational landslide is induced mainly by
hydrostatic pressure or confined water pressure caused by rainstorm (Kong and Chen, 1989;
Matjaž et al., 2004; Yin et al., 2005). The sliding body of the thick sandstone can slide along the





surface because of the combined action of the hydrostatic pressure in cracks and the uplift force of
the sliding surface (Wang et al., 1985; Zhang et al., 1994; Xu et al., 2006; Fan, 2007). At the same
time, the sliding soil, which is expanded by water, leads to a slip between nearly horizontal layers
(Yin et al., 2005). The other perspective is the hard rock layer covered by the upper layer, such as
granite and sandstone, has a crushing effect on the lower weak rock layer, thereby causing the
rock mass to expand laterally to form a landslide (Cruden et al., 1996; ЕМЕЛЬЯНОВА, 1986).
Regarding the theoretical study on rainfall-induced translational landslide, domestic and
foreign scholars have used physical simulation experiments, mechanical model analysis, and
satellite remote-sensing methods to investigate the genetic mechanism, initiation criteria, and
sensitive safety factors. Fan Xuanmei et al. (2008) reproduced the deformation and failure process
of landslides through physical simulation, and verified further the formation mechanism and
starting criterion formula of the flat-push landslide studied previously by Zhang et al. (1994).
Sergio et al. (2006) focused on the influence of pore-water pressure on the stability of
rainfall-induced landslides, and studied soil failure model based on pore-water pressure by
simulation experiment. Mario et al. (2008) and Teixeira et al. (2015) selected rainfall data from
historical heavy rainfall condition, and used physical experiments to establish an optimization
model for rainfall-induced landslide initiation criteria for landslides in the southern Apennines and
shallow landslides in northern Portugal; the researchers also evaluated landslide susceptibility and
safety factors to evaluate the possibility of landslide resurrection induced by rainstorm. Barlow et
al. (2003) and Martin et al. (2005) used US land satellite ETM+ and DEM data to detect the
residues of translational bedrock landslides in alpine terrain. Jessica et al. (2018) used resistivity
imaging to investigate the Montaguto translational landslide in the southern part of the Apennines;



the researchers also established a refined geometric model to observe the lithologic boundaries,
structural features, and lateral and longitudinal discontinuities associated with sliding surfaces.

Through the data collation and analysis of the current research status of the translational

landslide, domestic and foreign scholars have conducted further research on the formation
characteristics and genetic mechanism of translational landslides. Certain physical data models
have been established by using historical data on rainfall, and physical simulation experiments
have been conducted in the laboratory to verify the damage model. However, the actual
engineering cases of on-site monitoring for this type of landslide have not been observed in
domestic and foreign studies. Therefore, the research on translational landslide lacks monitoring
engineering and measured data on landslide physical parameters, such as trailing edge crack width,
real-time rainfall, pore-water pressure, and groundwater level. Thus, demonstrating and validating
the deformation and failure mode of the translational landslide in the theoretical analysis is
difficult.

Based on the formation mechanism of the translational landslide established by previous

studies, this paper combines the work results of the geological hazard investigation in the Ba river
basin of Qingba–Longnan mountain area. This study selected a typical and special translational
landslide (the Wobaoshi landslide) in the working area and adopted field survey, long-period
monitoring methods (February 2015 to July 2018) model analysis and numerical simulation.
Through the comprehensive analysis between the theoretical model calculation and the monitoring
data, the instability conditions and variation failure model of this translational landslide under the
influence of heavy rainfall are studied.





## 1.  Landslide Characteristics

### 1.1. General Situation of the Wobaoshi Landslide

The Wobaoshi landslide is located in the Ba river basin in the Qinba–Longnan mountainous area. Its specific location is in Baiyanwan village, Sanhui town, Enyang district in Bazhong city. Fig. 1 presents the geographical location and elevation information. The Wobaoshi landslide is located on the left bank of the Shilong river. The front edge of this landslide is in the curved section of the river, and the left boundary gully is located on the concave bank on the left bank of the river. The landslide area is classified as a red-bed layer in a low mountainous area, the vegetation of the sliding body is dense, and geomorphic unit is cuesta structural slope. The geological structure of the landslide body is in the south side of the Nanyangchang anticline, and the stratum is the mudstone and sandstone interbed of the Penglaizhen Formation of upper Jurassic series (Chen et al., 2015).

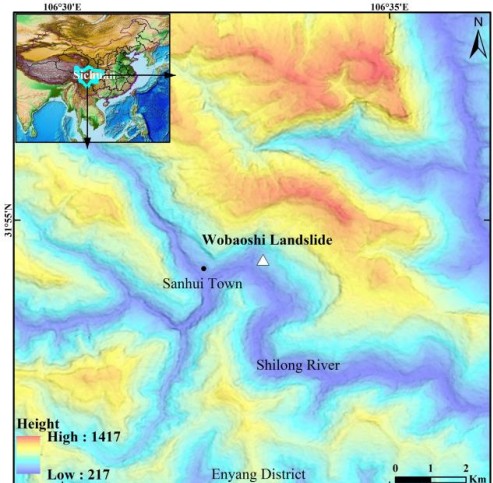

Fig. 1 Geographical location and elevation map of the Wobaoshi landslide.

This landslide is common in the eastern subtropical monsoon climate region, where the rainfall is abundant and mostly concentrated from May to October, accounting for 75%–85% of the total annual rainfall. The monthly average rainfall is above 100 mm, of which the highest is in July, and the monthly average in July is over 200 mm and often accompanied by rainstorm. The rainfall gradually decreases after August. The types of groundwater are mainly fissure water in




weathered bedrock and pore-water in trailing edge cracks, and the dynamic change of groundwater
is affected greatly by climatic change (Chen et al., 2015). The rapid immersion of groundwater
softens the joint surface of soil and rock formation, especially under heavy rainstorm, when the
groundwater level rises and the pore-water pressure increases sharply. This condition changes the
stress mode and equilibrium state of the rock and soil mass, thereby easily inducing a landslide.
**1.2. Landslide Characteristics and Forming Conditions**
1.2.1 Landslide Characteristics
According to the satellite remote-sensing interpretation and landslide survey, the shape of the
sliding body is a flat long rectangle on the plane. Its longitudinal (sliding) direction is nearly 32 m,
the lateral width is 160 m, the average thickness of the sliding body is approximately 30 m, and
the volume is approximately $1.536 \times 105$ m3. It belongs to small- to medium-sized landslides
according to the scale size. The sliding direction of the landslide is 249°, and the overall
occurrence of the rock formation is $170° – 180°\ \angle 6° – 8°$. The strike is nearly parallel to
the overall trend of the bank slope, which is a typical nearly horizontal consequent bedding rock
slope. Fig. 2 shows a planar graph of the Wobaoshi landslide and photographs of four observation
points. Fig. 3 presents I-I' sectional graph of the landslide.

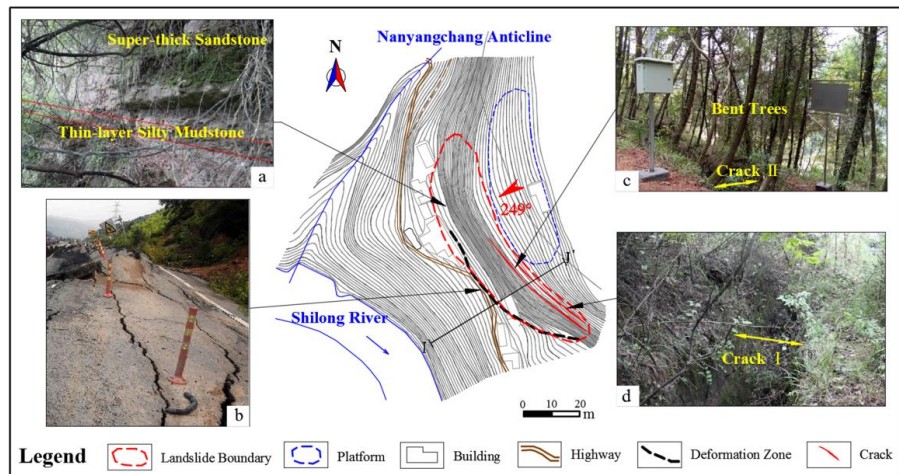

Fig. 2 Planar graph of the Wobaoshi landslide and photographs of observation points: (a) exposed bedrock in front
edge, (b) roadbed is pushed uplifted in front edge, (c) crack II and bent trees, and (d) crack I.





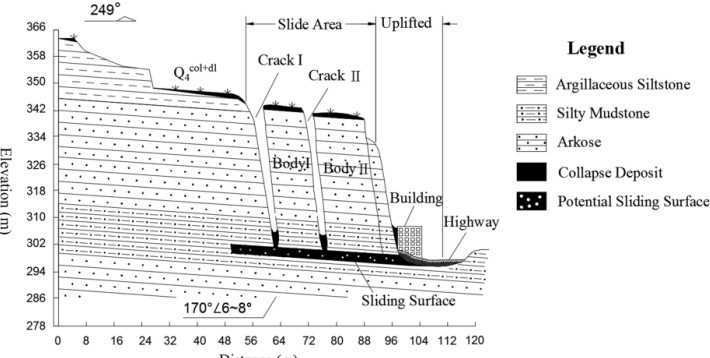

Fig. 3 I-Iˈ sectional graph of the landslide

As Fig. 2 shows, the landslide shape is special, the longitudinal length is much less than the
lateral width and even smaller than the thickness of the sliding body. Therefore, it can easily be
mistaken for a multi-stage dangerous rock mass with dumping deformation during the disaster
investigation. According to Fig. 3, the inclination of the landslide is almost erect, and a group of
long and straight structural planes that are parallel to the slope cuts the slope into two thin plates
(sliding bodies I and II), the surface structure of the slope has a certain degree of aperture, both
sides of the crack are closed, and the bottom of the crack is filled with clay with gravel and
collapse deposits.
1.2.2 Forming Conditions
The sliding body of the Wobaoshi landslide formed two obvious cracks from the outside to
the inside, which cut and disintegrated the sliding body into plate-shaped blocks from front to
back, as shown in the photographs of observation points c and d in Fig. 2. Then, the plate-shaped
sliding bodies I and II were formed. The landslide is a two-stage translational landslide in which
the longitudinal length of the sliding body I is 12 m, the identifiable lateral width is approximately
70 m and the thickness is approximately 30 m, the longitudinal length of the sliding body II is 16
m, the identifiable lateral width is approximately 65 m, and the thickness is approximately 28 m.
The sliding body I forms crack I with the trailing edge of the landslide, and the sliding body II
forms crack II with the sliding body I. When a large rainfall intensity occurs during the rainy
season, the pore-water in the cracks can be observed, thereby indicating that cracks I and II have
preferable water-storage conditions.

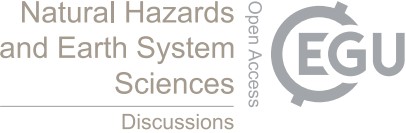

As the photo of observation point c in Fig. 2 shows, bent trees grow on the trailing edge of
the landslide bodies I and II. The trees on the landslide are skewed with the soil mass sliding, and
after the sliding stops, the upper part of the trunk turns to the upright state year by year. The
existence of bent trees represents the tendency of the slope body to become unstable or the
existing landslide accumulation body tends to slide again, and it is also the historical evidence of
the slow sliding of the landslide (Zhang Lizhan et al., 2015).
As the photo of observation point a in Fig. 2 shows, the shallow surface of the Wobaoshi
landslide is a 2–3 m thick layer of collapsed and plowed soil. The sliding body is composed of
extremely thick sandstone with good integrity, and the bottom sliding surface is a weak interlayer
consisting of silty mudstone. In summary, the Wobaoshi landslide is a typical and special
translational landslide, and according to the characteristics of its plate-shaped body, it can be
considered a plate-shaped landslide (Fan et al., 2008; Xu et al., 2009).
According to the characteristics of the Wobaoshi landslide, the formation conditions are
inferred; in other words, during the heavy rain, the group of open cracks parallel to the slope in the
sliding body is concentrated and quickly filled with water, and the sliding bodies I and II slide
horizontally along the contact surface of the bottom sand-mud rock weak layer. This condition
leads to the uplift of residential houses and highways in the front edge, as shown in the photo of
the observation point b in Fig. 2.

## 2. Landslide Monitoring Scheme and Monitoring Data Analysis

### 2.1. The Long-period Monitoring Scheme

According to the detailed investigation of the Wobaoshi landslide, two cracks extend through
the sliding surface at the trailing edge of the landslide, and the pore-water in the cracks exists in
the condition of heavy rain. As the hydrostatic pressure in the cracks strongly influences the
stability of the plate-shaped landslide (Fan Xuanmei et al., 2008; Guo Xiaoguang et al., 2013),
rainfall and pore-water pressure was conducted from February 2015 to July 2018 to determine the
landslide state in different periods such as rainy and non-rainy seasons, as well as the interaction
between multilevel plate girders and sliding surface, in the nearly three-and-a-half-year period of




182 long- period real-time monitoring of cracks. Fig. 4 shows the layout graph of the monitoring

183 equipment.

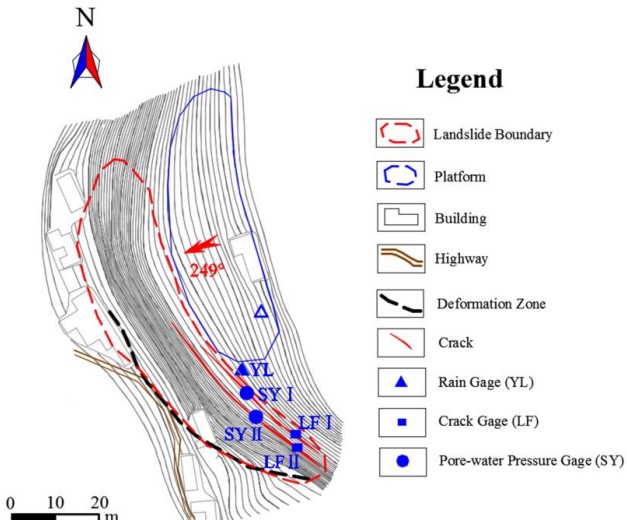

185           Fig. 4 Layout planar graph of the monitoring equipment

187   As Fig. 4 shows, two non-contact crack automatic monitors, LF I and II, are respectively

188 installed on both sides of cracks I and II, to record the real-time variation of the width of the two

189 cracks (Yimin Liu et al., 2015). An automatic rain gage is installed in flat space and no tree

190 occlusion is seen at the trailing edge of the landslide to record the real-time and cumulative values

191 of the rainfall. Two pore-water pressure gages are respectively installed at the bottom of crack I

192 and II to measure the pore-water pressure. The value of pore-water level, $h_c$, can be calculated by

193 the installation depth of the pore-water pressure gage, $h_i$, depth of the crack, $H$, and measured

194 value of the pore-water pressure gage, $h_m$, and $h_c = H - h_i + h_s$.

195   In this example, the initial width value of crack I is 5.640 m, and the initial width value of

196 crack II is 4.492 m (first measurement time is in January 2015); and the installation depth $h_{i1} =$

197 24.72 m, and the depth of crack I is $H_1 = 38$ m, and $h_{c1} = 13.28\text{m} + h_{m1}$; and the installation depth

198 $h_{i2} = 24.85$ m, and the depth of crack I is $H_2 = 35$ m, and $h_{c2} = 10.15\text{m} + h_{m2}$. The monitoring

199 frequency of the crack width is three times a day, the monitoring frequency of the pore-water

200 pressure is two times a day, and the rainfall intensity adopts the accumulative value of one month.

201 The multi-parameter monitoring data is transmitted to the monitoring server through the GPRS




network.

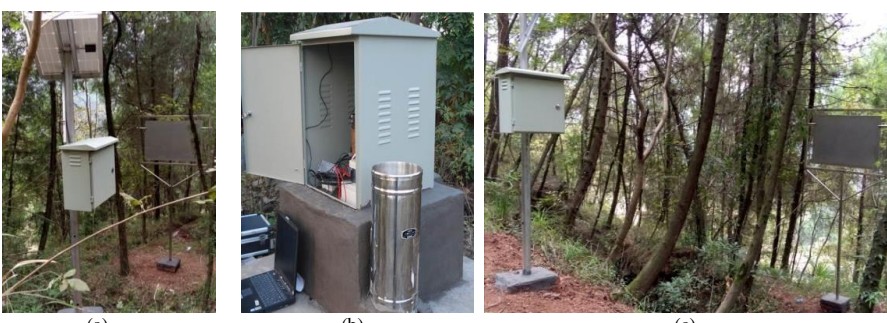

(a)                              (b)                                  (c)
Fig. 5 Photos of monitoring instrument installation: (a) Crack I gage; (b) Rain gage and pore-water pressure gage;
(c) Crack II gage.
**2.2. Monitoring Data Analysis**

Through the monitoring work on the Wobaoshi landslide for three-and-a-half years (February

2015 to July 2018), this study selects the typical data of the width of cracks I and II, the
pore-water pressure and rainfall intensity, details of this monitoring data are in attached Tables 1
and 2. The corresponding time curves in Fig. 6 show the monitoring data of the rainfall intensity
and the width of cracks I and II. Fig. 7(a) presents a comparison curve of the monitoring width
data of crack I and its pore-water pressure, and Fig. 7(b) presents a comparison curve of the
monitoring width data of crack II and its pore-water pressure.

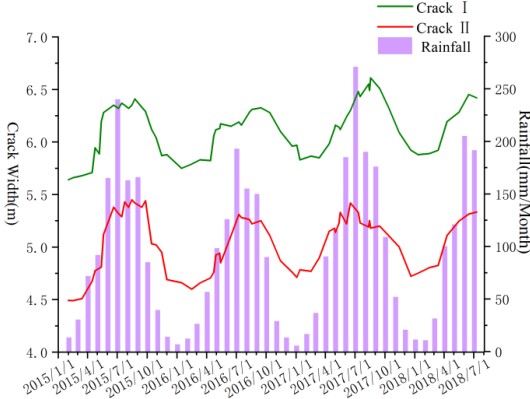




Fig. 6 Monitoring data curves (rainfall intensity and width of cracks I and II)


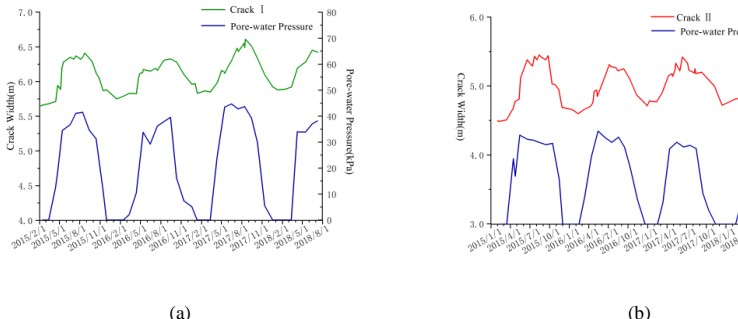


(a)                                              (b)

Fig. 7 Monitoring data curves: (a) width of crack I and its pore-water pressure; (b) width of crack II and its
pore-water pressure.


Based on the comprehensive comparison and analysis of the data curves in Figs. 6 and 7, this

study concludes that the Wobaoshi landslide is still in creep deformation state, and the
plate-shaped sliding body exhibits a regular trend with the change of rainfall intensity and
pore-water pressure. Crack I and crack II show a preferable water-storage capacity during the
rainy season, and the increase of the pore-water pressure affects the crack width variation. The
specific analysis is as follows:

(1) A clear correspondence exits between the absolute amount of crack width change and

season change (rainfall intensity), the magnitude of the rainfall intensity determines the change of
the width of the two cracks. As Fig. 6 shows, the width of cracks I and II widen as the rainfall
intensity increases during the rainy season (May to September), and the crack width gradually
shrinks as the rainfall intensity weakens during the non-rainy season (October to April in the next
year). As Fig. 8 indicates, the maximum width of crack I reaches 6.615 m, and the absolute
stretching amount of this crack is close to 1 m in July–August 2017 (monthly rainfall exceeding




250 mm). The maximum width of crack II is also in the range of 5.40–5.45 m, and the absolute
stretching amount is more than 1 m in July–August 2015 and July–August 2017. During the
non-rainy season, when the rainfall intensity weakens, the crack width begins to shrink and
decreases to a minimum in January of each year..

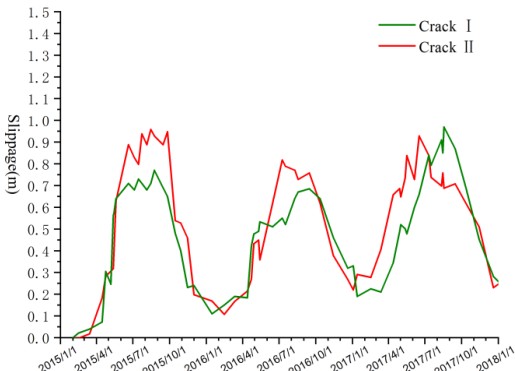


Fig. 8 Absolute slippage amount curves of crack I and II


(2) The width of cracks I and II tend to increase year by year, indicating that the two-stage

sliding body of the Wobaoshi landslide is still moving along the sliding surface due to the
influence of rainfall. In Fig. 6, the measured data in the monitoring period indicate that the
minimum width of crack I and II is gradually increasing, and the maximum value is affected
greatly by the rainfall intensity during a particular month.

(3) Fig. 7 shows that the stretching of crack I and crack II or both have the same tendency as

the pore-water pressure or that the magnitude of pore-water pressure determines the width
variation of the cracks. Fig. 7 also shows that the water-storage capacity of crack I is good during
the rainy season, and after the sliding body slides, it can maintain a certain pore-water level due to



rainfall replenishment. Meanwhile, the increase of rainfall intensity leads to the increase of water
level in the cracks, and the increase of pore-water pressure has a positive effect on the initiation of
the plate girder. The curve in Fig. 8 shows that the increase in pore-water pressure has a significant
causal relationship with the stretching of the cracks.

## 3.    Model Calculation and Numerical Simulation

### 3.1. Model Establishment and Stability Calculation

Aiming at the genetic model of the evolution process of the Wobaoshi landslide, the
mechanical model of the plate-shaped sliding body is established and the stability is calculated,
and combined with the monitoring data for comparative analysis. According to previous findings,
when many penetrating cracks are parallel to the slope in the rock mass, after the cracks are filled
with water at the same time, the water pressure on both sides of the plate-shaped body are
basically in a balanced state except for the outermost body. However, once the outer body slides,
due to the sudden decrease of the pore-water level in the trailing edge crack, the water pressure
immediately following the plate-shaped body becomes unbalanced, and new sliding damage is
generated (Fan, 2007; Xu, 2008). Therefore, for the failure mode of the two-stage plate girders of
the Wobaoshi landslide, this study selects a typical section of the plate-shaped sliding bodie and
establishes the mechanical model, as shown in Fig. 9. First, this section carries out stability
analysis of the outer layer plate girder II, and then analyzes the inner plate girder I.

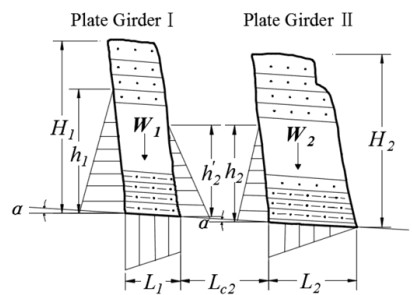






Fig. 9 Mechanical model of two-stage plate-shaped sliding bodies

Fig. 9 shows, $\alpha$ is the angle of the sliding surface, $h_1$ and $h_2$ are respectively heights of

pore-water level in crack I and II, $L_1$ and $L_2$ are respectively widths of plate girder I and II, $L_{c2}$ is

the distance between plate girder I and II, $H_1$ and $H_2$ are respectively heights of plate girder I and

II, and W1 and W2 are respectively the self-weights of the plate girder I and II per unit width.

According to the relationship between stability coefficient of plate girder, $K$, and the height of

pore-water level, $h$, in Fig. 8 (Zhang et al., 1994; Xu et al., 2010), and in consideration of the

internal cohesive force in sliding surface, the calculation formula of the stability coefficient $K_2$ of

the outer layer plate girder II is expressed as follows:

$$K_2 = \frac{\left( W_2 \cos\alpha - \frac{1}{2}\gamma_w h_2 L_2 - \frac{1}{2}\gamma_w h_2^2 \sin\alpha \right) \tan\theta + cL_2}{\frac{1}{2}\gamma_w h_2^2 \cos\alpha + W_2 \sin\alpha} \qquad (1)$$

In formula (1), $c$ is the internal cohesion in sliding surface, $\gamma_r$ is the saturated gravity of

sandstone, $\gamma_w$ is the gravity of water, and $W = H \cdot L \cdot \gamma_r$. $K_2$ is set to 1, that is, the plate girder II is set

in critical sliding state (GB/T 32864-2016, 2017), and calculation formula (2) of the maximum

pore-water level of the plate girder II, $h_{cr2}$, is derived by formula (1).

$$h_{cr2} \approx \frac{1}{2\cos\alpha}\left[ L_2^2 \tan^2\theta + \frac{8}{\gamma_w}\left( W_2 \cos\alpha \tan\theta - W_2 \sin\alpha + cL_2 \right)\cos\alpha \right]^{\frac{1}{2}} \qquad (2)$$
$$-\frac{L_2}{2\cos\alpha}\tan\theta$$

According to the triaxial confining pressure experimental data of rock core of the Wobaoshi

landslide (Chen et al., 2015), the internal friction angle of the sliding surface $\theta = 11.2°$, the

saturated gravity of sandstone $\gamma_r = 19.2$ kN/m³, the gravity of clear water $\gamma_w = 9.8$ kN/m³, and the

internal cohesion of the sliding surface $c = 10.2$ kPa. According to the sectional graph of the



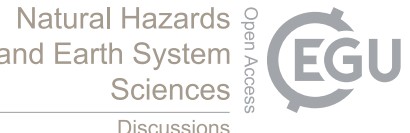

Wobaoshi landslide (see Fig. 2), $H = 35$ m, $L = 16$ m, $\alpha = 6\,°$. Therefore, according to formula (2),
$h_{cr2}$=13.896m.

On the basis of stability analysis of the plate girder II, combines with formula (1), formula (2)

and Fig. 7, the calculation formula of the stability coefficient $K_1$ of the inner layer plate girder I is
expressed as formula (3). And $h_2^{'} = h_2 - L_{c2} sin\alpha$, $L_{c2} = 3.8$m, therefore, $h_2^{'} = 13.499$m.

$$K_1 = \frac{\left[ W_1 \cos\alpha - \frac{1}{2}\gamma_w \left( h_1 + h_2^{'} \right) L_1 - \frac{1}{2}\gamma_w \left( h_1^2 - h_2^{'2} \right) \sin\alpha \right] \tan\theta + cL_1}{\frac{1}{2}\gamma_w \left( h_1^2 - h_2^{'2} \right) \cos\alpha + W_1 \sin\alpha} \tag{3}$$

Similarly, $K_1$ is set to 1, and in the plate girder I, $H_1 = 38$m, $L_1 = 12$m, $\alpha = 6\,°$, $h_2^{'} = 13.499$m,

therefore, the maximum pore-water level $h_{cr1}$ of the plate girder I can be calculated by using the
formula (3), and $h_{cr1}$=17.249m.

The preceding calculation results show that when the pore-water level at the trailing edge of

the plate girder reaches the maximum height at which the landslide starts, that is, when the
$h_{cr1}$=17.249m, $h_{cr2}$=13.896m, the pore-water pressure triggers the plate-shaped sliding bodies. The
next section aims to verify the pore-water monitoring data, which are acquired by the landslide
monitoring.

The pore-water monitoring data in Section 2.2, which is acquired by the landslide monitoring

engineering is used to test the calculating formula of the maximum height of multi-stage plate
girders, $h_{cr}$. According to the monitoring data of pore-water pressure and installation depth of the
sensors, the actual maximum height value $h_{c1}$ and $h_{c2}$ of the pore-water level have been calculated
in attached Table 3. Combined with the change of the absolute stretching amount in Fig. 8, the



typical data of the measured pore-water level is selected, which corresponding to sudden change
of the absolute slippage (see Table 3 for details), as shown in Figure 10.

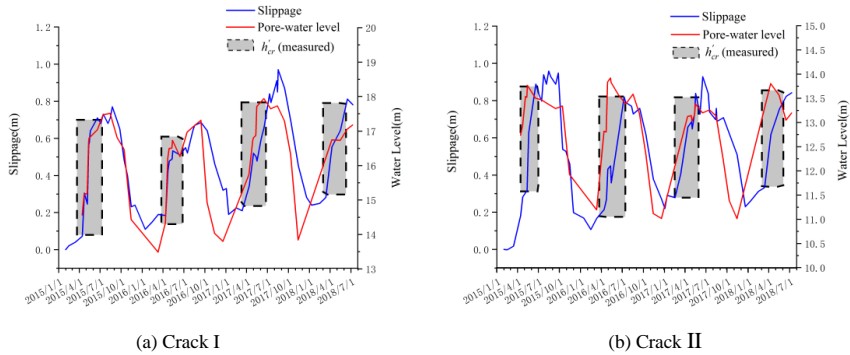


(a) Crack I                                            (b) Crack II

Fig. 10 Determination of the maximum pore-water level $h_{cr}^{'}$(measured)


The dotted box in Fig. 10 represents the value of pore-water level when the sliding body is

sliding, that is, the maximum pore-water level, $h_{cr}^{'}$ , which causes the sliding body to be unstable.
Then compare the $h_{cr}^{'}$ (measured) in Fig. 10 with the relationship between the pore-water level, $h$,
and stability coefficient of the plate girder, $K$, in formula (1) and formula (3), which is shown in
Fig. 10.

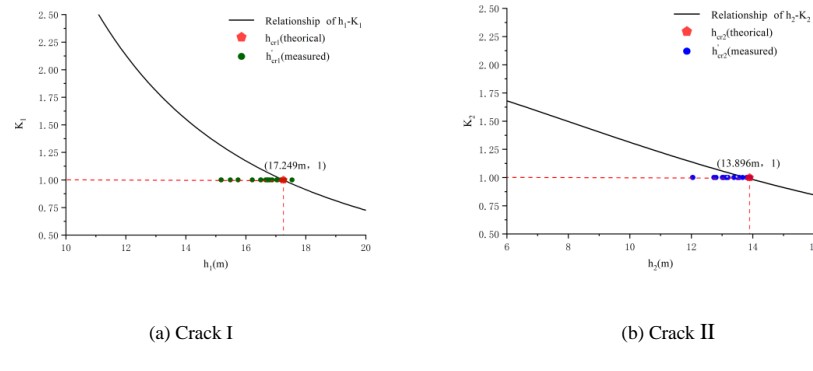


(a) Crack I                                            (b) Crack II

Fig. 11 Comparison figure of $h_{cr}^{'}$(measured) and $h_{cr}$ (theorical)




In Fig. 11, the curves of relationship of h-k represent formula (1) and formula (3). The
frequency of $h'_{cr}$(measured) in Fig. 11 shows that most of the monitoring pore-water levels are not
bigger than the theoretical calculations. The Wobaoshi landslide monitoring example shows that in
most cases, when $h'_{cr}$(measured)$\leqslant h_{cr}$ (theorical), the pore-water pressure will cause the instability
of the sliding body.
**3.2. Numerical Simulation of the Plate-shaped Sliding Bodies**
The numerical simulation and calculation of the plate girder is carried out by MIDAS GTS
NX geotechnical finite element software. Firstly, the 1:1 sliding body model in Fig. 9 is
introduced into the finite element software, and mechanical parameters of the sliding body model
are shown in Table 4. The boundary conditions are set as follows:
(1) Displacement boundary: the left and right boundaries constrain the X direction
displacement, TX = 0; the bottom boundary: constrain the X and Y direction displacement TX =
TY = 0;
(2) Seepage conditions: set the water level at the left and right boundaries to be 342m and
275m respectively.
The typical data of pore-water level in Table 3 is introduced into the finite element model,
and then numerical calculations are performed to obtain typical deformation and displacement
states of the plate-shaped sliding bodies in the rainy season and non-rainy season, as shown in Fig.

12.




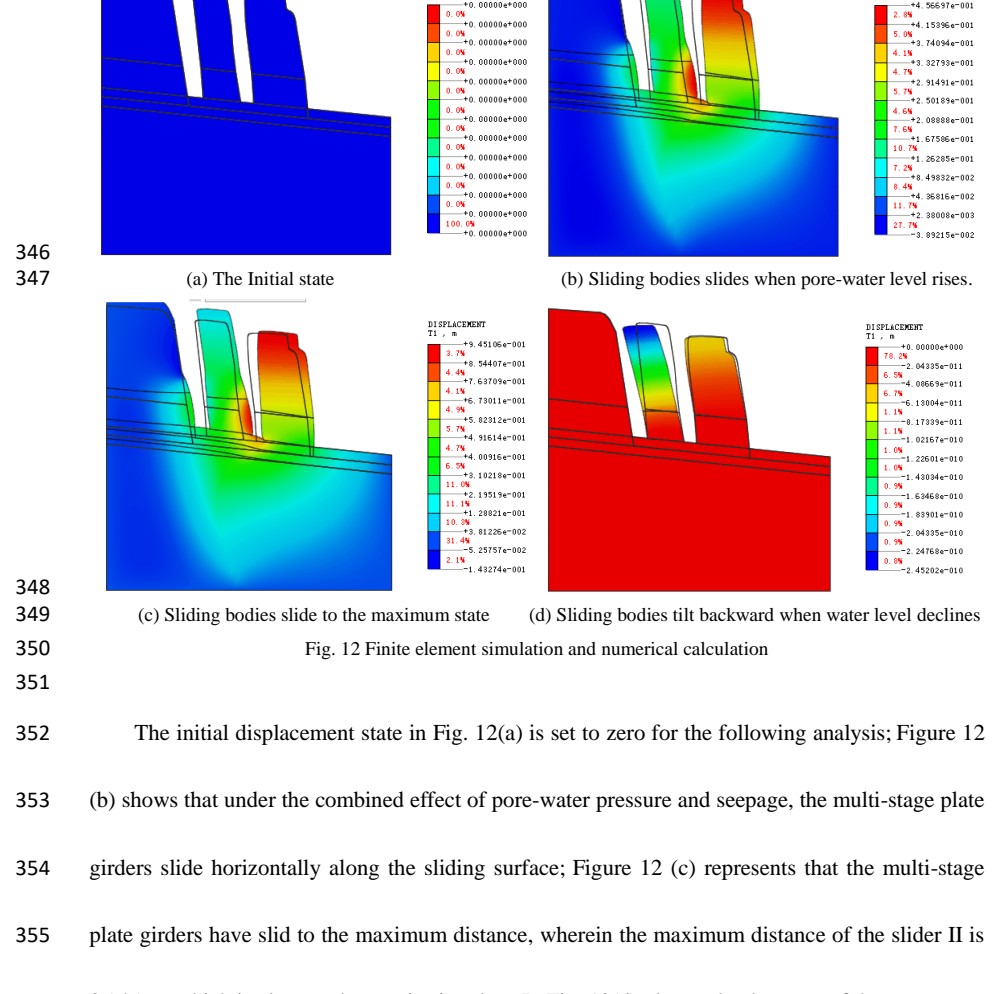

(a) The Initial state                          (b) Sliding bodies slides when pore-water level rises.

(c) Sliding bodies slide to the maximum state      (d) Sliding bodies tilt backward when water level declines
Fig. 12 Finite element simulation and numerical calculation

The initial displacement state in Fig. 12(a) is set to zero for the following analysis; Figure 12

(b) shows that under the combined effect of pore-water pressure and seepage, the multi-stage plate
girders slide horizontally along the sliding surface; Figure 12 (c) represents that the multi-stage
plate girders have slid to the maximum distance, wherein the maximum distance of the slider II is
0.945 m, which is close to the monitoring data; In Fig. 12(d), due to the decrease of the pore-water
level in the non-rainy season, the sliding bodies I and II have a same tendency to tilt backward.
The calculation results of the numerical simulation can corroborate with sliding body mechanics
model and landslide monitoring data.

## 4.  Discussion

As mentioned in the previous sections, this special type of translational landslide, which has a



plate-shaped sliding body and is generally formed in an extremely thick sandstone slope with a
thin cover layer, is nearly horizontal and has good integrity. According to the traditional theory of
granular equilibrium limit, deformation or sliding movement of this nearly horizontal bedrock
slope is almost impossible, and the likelihood of forming a landslide is minimal. However, in the
investigation of geological hazard hidden dangers, a special structure of translational landslide
occurs in the red-bed zone of the Qinba–Longnan mountainous area. Owing to the dense
population and large infrastructure in the working area, the plate-shaped landslide is characterized
by large volume, concealment, and sudden and strong destructive ability. The collapse is often
considered to be small-scale and its danger is ignored. Therefore, in the investigation and risk
assessment of geological hazards, the characteristics of the plate-shaped landslide and the
deformation and failure mode should be combined to detect the hidden dangers with the
geological conditions of the landslide. Combines the results of predecessors, discussion of this
paper is shown in the following three aspects.
**4.1. Deformation and Failure Mode Exploration of the Wobaoshi Landslide**

The monitoring results of the Wobaoshi landslide in this case validate the rainfall-triggered

failure mode of the translational landslide (Zhang Yuyuan et al., 1994), According to the landslide
monitoring data and the numerical simulation of the plate-shaped sliding bodies, the deformation
and failure mode of the landslide is obtained, which is shown in Fig. 13. Fig. 13 shows
deformation of the plate-shaped sliding body of the Wobaoshi landslide during a monitoring
period (non-rainy season–rainy season–non-rainy season). As shown in Fig. 13(b), a large amount
of rainfall causes the cracks to be filled with water in rainy season, when the pore-water level
reaches the maximum height at which the landslide starts, increased pore-water pressure has a
positive effect on the initiation of the plate-shaped sliding body (Fan Xuanmei et al., 2007). When
the pore-water pressure rises to the threshold value, the plate-shaped landslide can be triggered. In
this monitoring case, the pore-water pressure can push the plate-shaped sliding body by nearly 1 m,
thereby resulting in the uplift of residential houses and highways in the leading edge. Therefore,
we can infer that one or more penetrability cracks should be parallel to the slope in the landslide
body. When the rainy season is approaching, the plate-shaped sliding body II begins to slide first,



and the water pressure balance in the cracks is destabilized. This condition causes the gliding of
the plate-shaped sliding body I, thereby forming a multi-stage translational landslide with the
characteristic of step-by-step backward movement.

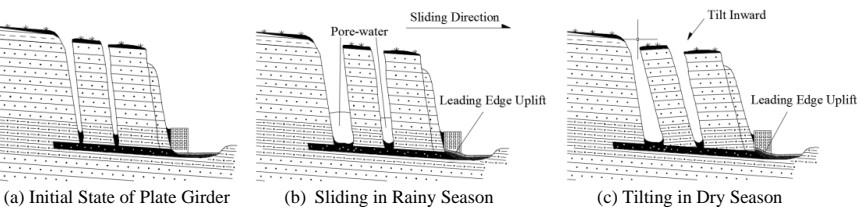

(a) Initial State of Plate Girder    (b) Sliding in Rainy Season    (c) Tilting in Dry Season


Fig. 13 Schematic of deformation and failure mode of the Wobaoshi landslide

As shown in Fig. 13(c), the plate girder is tilted to the trailing edge by the lower pore-water
level and its own weight with less rain during the non-rainy season, thereby causing the plate
girder to fall backward (inside the slope) until the top of the plate girder is in contact with the
slope surface, the crack width begins to shrink, and a narrow A-shaped crack is formed.
Monitoring data of the Wobaoshi landslide and numerical simulation of plate-shaped sliding body
also verify the deformation and failure mode of the plate-shaped landslide after occurrence (Xu et
al., 2010). Year after year, the cracks at the bottom of the slab-shaped sliding body grow larger,
and the degree of inclination of the plate girder continues to increase. The degree of arching of the
front edge also increases, which causes the stability of the landslide to decrease continuously,
thereby posing a high risk for the houses and roads on the front edge of the landslide.
**4.2. Determination of Maximum Pore-water Level $h_{cr}$**
The theoretical analysis and stability calculation of the mechanical model of the plate girder
is described in Section 4.1, along with the starting criterion for multi-stage sliding bodies of
translational landslide, that is, determination of the maximum water height in the crack, $h_{cr}$,
(Zhang et al., 1994) and calculation of value of the stability coefficient of the sliding body, $K$, (Xu
et al., 2010), which is determined by the theoretical calculation of strata inclination, shape, weight,
and physical properties (such as saturated gravity, $\gamma_r$, internal cohesion of the sliding surface, $c$,
and internal friction angle of the sliding surface, $\theta$) based on the limit equilibrium theory (Lin et



al., 2010). Therefore, the stability coefficient of the landslide decreases exponentially with the
increase of the water-filling height of the trailing edge crack (Fan, 2008; Xu et al., 2010).

In this case, the formula for calculating the maximum pore-water level, $h_{cr}$, deduced in

Section 3.1, comparing the measured data of the Wobaoshi landslide in Secion 2.2, we can observe
that the measured maximum pore-water level, $h_{cr}^{'}$, is close to the theoretical maximum pore-water
level, $h_{cr}$, thus verifying the correctness of calculation formula of $h_{cr}$, and instability conditions of
the sliding bodies. And the most measured data are slightly smaller than the theoretical calculation
value, that is, $h_{cr}^{'} \leq h_{cr}$. In other words, compared with the calculation formula of the maximum
water height proposed by Zhang et al. (1994) and the physical simulation experiment conducted
by Fan et al. (2008), the monitoring case of the Wobaoshi landslide shows that the measured data
$h_{cr}^{'}$ is mostly lower than the theoretical calculated value, $h_{cr}$, which can cause the instability of the
sliding body. The reason for the instability may be that the actual cohesion value $c'$ of the
sand-shale contact surface is smaller than the cohesive force value $c$ of the sliding surface in
formula (2) during the creep state of the landslide for a long time, or the frictional angle of the
sliding surface, $\theta$, changes slightly. According to the calculation of the stability coefficient, $K$, in
formula (2), when $c' \leq c$, $h_{cr}^{'} \leq h_{cr}$ is obtained, the plate girder slides in case of $h_{cr}^{'}$ (measured) is
not larger than $h_{cr}$ (theoretical).
**4.3. Optimization Methods of Landslide Monitoring**

Focusing on the plate-shaped translational landslide through the existing field monitoring

result experience and deformation and failure mode exploration, this study proposes the following
suitable monitoring methods for this type of landslide. First, long- period monitoring should be
conducted to obtain sufficient monitoring data, which mainly includes obtaining groundwater
level, pore-water pressure, rainfall intensity, and displacement data on the front edge of the
landslide during the rainy season, as well as focusing on the change of overall inclination of the
plate girder during the non-rainy season. The reason is the inclination angle α relative to the




sliding surface also changes after the sliding of the plate girder. Thus, the inclination measuring
device should be installed in the sliding body, in order to verify the theoretical exploration of
deformation mode of the plate-shaped sliding body in non-rainy season in Fig. 13(c). Furthermore,
a sensitivity analysis of various parameters affecting the stability coefficient K of the sliding body
(such as the pore-water level, internal cohesive force in saturated water, internal friction angle of
the sliding surface, and inclination angle of the plate girder) should be conducted on the basis of
the monitoring data. Therefore, it is beneficial to in-depth analysis and exploration of the
deformation and failure mode of the plate-shaped landslide and improves the success rate of
landslide warning.

## 5. Conclusions

Taking the case of the Wobaoshi landslide as an example, this study uses research methods

such as field exploration, a long- period monitoring engineering, mechanical model analysis and
numerical simulation, and to deeply analyze the instability conditions and failure characteristics of
a special type of translational landslide. The research results are beneficial to the stability analysis
and evaluation of this type of landslide. Targeted monitoring methods are proposed to enrich
theoretical research of the translational landslide. The following conclusions are drawn:

(1) The characteristics, formation conditions, and occurrence mechanism of rainfall-triggered

translational plate-shaped landslide are summarized. This type of landslide generally exists in a
consequent slope with the inclination angle of the sliding surface less than 10 °, a group of long
and straight structural planes parallel to the slope cuts the slope into several thin plates. The
plate-shaped sliding body generally consists of extremely thick sandstone, which is nearly



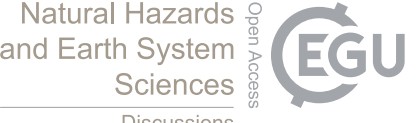

horizontal and has good integrity. The bottom sliding zone is a weak mudstone interlayer affected
by heavy rainfall, single-stage or multi-stage plate-shaped sliding bodies slide horizontally along
the bottom mudstone sliding zone.
(2) Based on establishment of a mechanical model of plate-shaped sliding bodies, the
relationship between stability coefficient of the multi-stage sliding body, $K$, and the pore-water
level, $h$, are obtained, and the maximum pore-water level, $h_{cr}$, which causes the instability of
multi-stage plate girders are calculated. The instability conditions of the plate-shaped sliding
bodies are also determined.
(3) Theoretical conclusions of the plate-shaped landslide research are verified by the
long-period monitoring data. The multi-parameter monitoring data show that the stability of the
sliding body is affected greatly by the rainfall intensity and pore-water pressure. The pore-water
pressure in the crack is positive for the beginning of the plate-shaped sliding body, which
demonstrates the rainfall-triggered failure mode of the translational landslide. This study compares
and analyzes the measured maximum pore-water level $h_{cr}^{'}$ and theoretical calculated value $h_{cr}$, and
discusses the influence of the change of internal cohesive force and internal friction angle on the
stability coefficient of the sliding body.
(4) Combined with landslide numerical simulation, this paper analyzes and explores the
deformation and failure modes of the plate-shaped landslide, that is,combined with the pore-water
pressure in the crack and seepage effect in the rainy season, the sliding bodies will slide
horizontally along the contact surface of the bottom sand-mud rock weak layer. During the
non-rainy season, the pore-water pressure decreases and disappears, the sliding body, due to its




dead weight, will be inclined to the trailing edge. On this basis, this paper proposes an
optimization monitoring methods to closely monitor the pore-water pressure, rainfall, and
landslide frontal displacement during the rainy season, and this method focuses on the overall
inclination angle change of the plate girder during the non-rainy season.

## Acknowledgments

We thank Dr. Long Chen at the Institute of Exploration Technology of CAGS for providing
landslide monitoring data. This work was supported by the National Natural Science Youth
Foundation of China (41804089), Project of Observation Instrument Development for Integrated
Geophysical Field of China Mainland (Y201802), and CGS of China Geological Survey Project
(1212011220169 and 12120113011100).

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

Chinese)





**Figure Captions**

Fig. 1. Geographical location and elevation map of the Wobaoshi landslide.

Fig. 2. Planar graph of the Wobaoshi landslide and photographs of observation points:

(a) exposed bedrock in front edge, (b) roadbed is pushed uplifted in front edge, (c)

crack II and bent trees, and (d) crack I.

Fig. 3. I-I' sectional graph of the landslide.

Fig. 4. Layout planar graph of the monitoring equipment.

Fig. 5. Photos of monitoring instrument installation: (a) Crack I gage; (b) Rain gage

and pore-water pressure gage; (c) Crack II gage.

Fig. 6. Monitoring data curves (rainfall intensity and width of cracks I and II).

Fig. 7. Monitoring data curves: (a) width of crack I and its pore-water pressure; (b)

width of crack II and its pore-water pressure.

Fig. 8. Absolute stretching amount curves of crack I and II.

Fig. 9. Mechanical model of two-stage plate-shaped sliding bodies.

Fig. 10 Determination of the maximum pore-water level $h_{cr}^{'}$(measured) .

Fig. 11. Comparison figure of $h_{cr}^{'}$ (measured) and $h_{cr}$ (theorical).

Fig. 12 Finite element simulation and numerical calculation.

Fig. 13 Schematic of deformation and failure mode of the Wobaoshi landslide.



**Table**

Table 1    Typical monitoring data of the Wobaoshi landslide

| Measured time | Crack I width (m) | Crack II width (m) | Crack I Pore-water pressure (kPa) | Crack II Pore-water pressure (kPa) |
|---|---|---|---|---|
| 2015/2/1 | 5.640 | 4.492 | 0 | 0 |
| 2015/4/24 | 5.945 | 4.774 | 18.561 | 27.303 |
| 2015/5/7 | 5.886 | 4.798 | 18.649 | 33.212 |
| 2015/5/13 | 6.203 | 4.810 | 33.134 | 33.036 |
| 2015/5/15 | 6.215 | 4.899 | 34.476 | 35.456 |
| 2015/8/15 | 6.350 | 5.451 | 41.474 | 31.625 |
| 2015/9/14 | 6.330 | 5.380 | 34.594 | 30.772 |
| 2015/11/15 | 5.871 | 4.952 | 11.280 | 17.395 |
| 2016/2/15 | 5.790 | 4.599 | 0 | 0 |
| 2016/4/13 | 5.824 | 4.706 | 10.378 | 26.156 |
| 2016/5/14 | 6.173 | 4.850 | 33.810 | 36.035 |
| 2016/7/17 | 6.161 | 5.281 | 36.162 | 31.664 |
| 2016/8/18 | 6.310 | 5.220 | 38.024 | 33.683 |
| 2016/9/15 | 6.325 | 5.251 | 39.298 | 29.723 |
| 2016/12/20 | 5.960 | 4.763 | 5.106 | 0 |
| 2017/2/16 | 5.865 | 4.770 | 0 | 0 |
| 2017/4/13 | 5.984 | 5.152 | 24.108 | 29.155 |
| 2017/5/17 | 6.118 | 5.332 | 43.463 | 31.703 |
| 2017/7/17 | 6.433 | 5.239 | 42.787 | 30.478 |
| 2017/8/15 | 6.490 | 5.255 | 43.639 | 29.273 |
| 2017/11/14 | 6.091 | 5.004 | 5.488 | 8.428 |
| 2017/12/20 | 5.922 | 4.723 | 0 | 0 |
| 2018/1/11 | 5.881 | 4.751 | 0 | 0 |
| 2018/4/10 | 6.194 | 5.110 | 33.957 | 35.819 |
| 2018/5/17 | 6.283 | 5.246 | 33.830 | 33.438 |
| 2018/6/16 | 6.452 | 5.315 | 36.995 | 28.391 |
| 2018/7/10 | 6.421 | 5.310 | 38.171 | 29.841 |



Table 2    Rainfall intensity value of the Wobaoshi landslide (mm/month)

| Year \ Month | 1 | 2 | 3 | 4 | 5 | 6 | 7 | 8 | 9 | 10 | 11 | 12 | Total |
|---|---|---|---|---|---|---|---|---|---|---|---|---|---|
| 2015 | | 13.5 | 30.5 | 71.8 | 121.9 | 165.0 | 240.1 | 163.0 | 166.1 | 85.0 | 39.6 | 14.1 | 1110.6 |
| 2016 | 6.9 | 12.5 | 26.5 | 56.8 | 98.4 | 126.1 | 193.2 | 155.1 | 150.0 | 90.3 | 29.1 | 13.5 | 958.4 |
| 2017 | 5.7 | 16.8 | 36.8 | 90.5 | 115.6 | 185.1 | 271.3 | 190.0 | 176.2 | 109 | 52.1 | 20.8 | 1269.9 |
| 2018 年 | 11.5 | 10.9 | 31.5 | 99.9 | 121.0 | 205.1 | 191.6 | | | | | | 671.5 |






Table 3     Measured pore-water level data of the sliding bodies

| Measured time | Crack Ⅰ slippage (m) | Measured pore-water level (m) | Crack Ⅱ slippage (m) | Measured pore-water level (m) |
|---|---|---|---|---|
| 2015/4/15 | 0.072 | 14.566 | 0.183 | 12.736 |
| 2015/4/24 | 0.305 | 15.174 | 0.282 | 12.936 |
| 2015/5/7 | 0.246 | 15.183 | 0.306 | 13.539 |
| 2015/5/13 | 0.561 | 16.661 | 0.318 | 13.521 |
| 2015/5/15 | 0.573 | 16.798 | 0.407 | 13.768 |
| 2015/6/20 | 0.711 | 17.032 | 0.888 | 13.502 |
| 2015/7/17 | 0.519 | 17.474 | 0.798 | 13.471 |
| 2015/10/16 | 0.481 | 16.470 | 0.538 | 13.340 |
| 2015/11/15 | 0.229 | 14.431 | 0.458 | 11.925 |
| 2016/1/15 | 0.108 | \ | 0.169 | \ |
| 2016/4/13 | 0.184 | 13.490 | 0.214 | 12.819 |
| 2016/4/23 | 0.421 | 14.339 | 0.269 | 12.804 |
| 2016/4/29 | 0.475 | 16.214 | 0.432 | 13.835 |
| 2016/5/11 | 0.469 | 16.494 | 0.449 | 13.920 |
| 2016/5/14 | 0.531 | 16.505 | 0.358 | 13.827 |
| 2016/6/15 | 0.508 | 16.731 | 0.618 | 13.574 |
| 2016/9/15 | 0.683 | 17.312 | 0.758 | 13.183 |
| 2016/10/12 | 0.637 | 14.930 | 0.618 | 12.360 |
| 2017/2/16 | 0.223 | \ | 0.278 | \ |
| 2017/4/13 | 0.344 | 15.741 | 0.658 | 13.125 |
| 2017/4/29 | 0.489 | 16.712 | 0.686 | 13.141 |
| 2017/5/2 | 0.518 | 16.799 | 0.648 | 13.024 |
| 2017/5/13 | 0.501 | 16.877 | 0.734 | 13.161 |
| 2017/5/17 | 0.476 | 17.715 | 0.838 | 13.385 |
| 2017/8/15 | 0.848 | 17.733 | 0.758 | 13.137 |
| 2017/9/16 | 0.869 | 16.324 | 0.333 | 12.235 |
| 2018/3/14 | 0.281 | \ | 0.618 | 11.013 |
| 2018/4/10 | 0.552 | 16.745 | 0.754 | 13.805 |
| 2018/5/17 | 0.643 | 16.732 | 0.333 | 13.562 |


594            Table 4     Mechanical parameters of sliding body model

| Lithology | Elastic Modulus (N/m²) | Poisson Ratio | Gravity (N) | Internal Cohesion (N/m²) | Internal Friction Angle | Permeability Coefficient (cm/s) |
|---|---|---|---|---|---|---|
| Arkose | 600000 | 0.25 | 19200 | 30000 | 36° | 1.20E-07 |
| Silty Mudstone | 360000 | 0.28 | 19000 | 20000 | 30° | 6.00E-07 |
| Clay | 300000 | 0.3 | 18000 | 10200 | 11.2° | 1.20E-06 |
