# Peer review of "Analysis of instability conditions and failure mode of a special"

_Natural Hazards and Earth System Sciences, 2019_

## Referee Comment (RC1) · Anonymous Referee #1 · 5 Jul 2019

Submitted paper describes very interesting and specific natural hazard phenomena. Extensive work was done, and a lot of significant data were gathered during the long-term geotechnical landslide monitoring. Although the topic is suitable for publication, improvement of the current manuscript is needed. I suggest reconstructing and rewriting of individual chapters. Language, in general is bad and some sentences are unclear. Some professional terms are incorrect. Proofreading of the final version is necessary. In the Introduction I expect a little more information (what, where, why) and presentation of the main aim of the paper. Does the landslide threaten something?

[Figure]

Why it is important to be studied? Introduction describes several detail characteristics of landslide which would be more suitable for the chapter - Results. In the Landslide characteristics the geological structure is not clearly presented (i.e. not clearly summarized after Chen et al., 2015). Some other: line 122: "overall occurrence of the rock formation" = "dip-and-strike of beds", line 128: "planar graph" -> "topographic map"; Fig. 3: arkose -> sandstone in the text; "rain gage -> rain gauge" etc. Monitoring results and numerical modelling results are clearly presented (graphs, models). Chapter Discussion is clearly written. The system of long-term gradual opening of the cracks and short-term opening and closing of cracks (blocks tilting) (very characteristic on the graphs) could be described more in detail. Any possible mitigation measures or additional monitoring suggested? However, I support the publication of this interesting topic and advise the major revision of the manuscript.

---

## Short Comment (SC1) · 24 Jul 2019

NHESS discussion on paper by Y. Liu, C. Wang, G. Gao, P. Wang, Z. Hou, Q. Jiao

The Authors discuss in their paper devoted to ÂńÂǎinstability conditions and failure mode of a translational landslideÂǎÂż a case study, whose main interest lies on its careful long term monitoring. The characteristics of this landslide are remarkably described and these results will certainly constitute a very valuable data bank for future studies. Moreover the basic mechanical analysis of the related boundary value prob-

lem has been conducted in a very convincing manner and the failure mechanism is clearly exhibited. So all the ingredients have been collected to form a perfect basis for the modelling. Let us insist here on the fact that a numerical or analytical mechanical model of natural hazards - to be fruitful - has to take into account as properly as possible the discriminant or critical features, exhibited by preliminary geological/mechanical/pluviometric analyses. These analyses have to be based on a well designed monitoring campaign and the issueing data bank. This is clearly the case here and the Authors have to be congratulated for having achieved that – particularly the correlation between rainfall intensity and block motion.

So what is interesting to be discussed now is the choice of the modelling strategy. The analytical model is based on limit equilibrium method, whose limits have to be recalledĂă: - this is a purely static method, - it ignores the influence of strain history, while it is well recognised that soils present an important hardening regime involving large plastic strains before failure, - a coulombian friction is taken into account, while the soil can fail before reaching Mohr-Coulomb plastic limit condition according to the second order work criterion, which is a more general material instability criterion. This is particularly important here, since the sliding surface has a very low slope angle of 6-8 degrees. Thus the instabilities are certainly appearing before Mohr-Coulomb criterion. - the hydro-mechanical constitutive relation of the sliding zone, which plays here a central role, can not be taken into account.

Having in mind these drastic limits, the reader is waiting for a deeper and more realistic finite element computation. However the finite element modelling is presented with very few details. According to the values of parameters as given on table 4, it seems that all involved geomaterials ( rocks and soils) have been considered as associate plastic materials. Thus their dilatancy angle is assumed to be equal to their friction angle, what is clearly not verified experimentally. Let us note that the constitutive elasto-plastic matrix is symmetric for associate materials, preventing to describe all instabilities and bifurcations occuring before the Mohr-Coulomb plastic limit surface.This is probably the

reason why it has been necessary to choose a so low friction angle of 11.2° for clay –
what seems unrealistic.

---

## Author Comment (AC1) · 24 Jul 2019

We are very grateful to this referee comments, and we have carefully read and considered the referee's comments, and these comments are important for improving the quality of this manuscript. Based on these comments, we have made carefully modification and proofreading on the original manuscript, the detail modifications are mainly for the chapter - Abstract and Introduction, which have been reconstructed and rewrited, and description of the Wobaoshi Landslide's geological structure and characteristics have been modified and reconstructed, and language, grammar and professional

terms in English also have been improved. Please see the detailed revision, and we carefully proof-read the manuscript to minimize typographical, grammatical, and bibliographical errors, and the modified parts are marked in red in the revised manusript. Thank you very much for your suggestion and consideration, and we look forward to hearing from you. Best regards, Yimin Liu, Chenghu Wang and Pu wang.

Detailed revision: 1. Language, in general is not good and some sentences are unclear. Some professional terms are incorrect. Modification: For the unclear parts and unprofessional terms in the manuscript, we have made serious changes in English expression to ensure clearly presented, and proofreading also has been done, and the revised changes are in red in the manuscript. For instance, "overall occurrence of the rock formation" has changed to "overall attitude of rock bed" in line 122, "planar graph" has changed to "topographic map" in Fig. 3, "gage" has changed to "gauge", "long-period" has changed to "long-term", "heavy rain" has changed to "periodic rain", "rainy season" has changed to "monsoon", "mechanical" has changed to "geomechanical", "field exploration" has changed to "field investigation", "starting" has changed to "initiation", "theorical" has changed to "theoretical" in the manuscript, and we have spelled all abbreviations and acronyms at their first mention in the main text, such as enhanced thematic mapper called ETM+, and digital elevation model called DEM in line 67 and 68. Thus, these descriptions will be more accurate and professional than before.

2. Introduction, there is a little more information (what, where, why) and presentation of the main aim of the paper. Does the landslide threaten something? Why it is important to be studied? Modification: This comment is very important for the paper, and the presentation of the main aim in original manuscript is lack of refinement and detailed narrative. Therefore, we modify and supplement the purpose and importance of this manuscript, firstly, the occurrence of plate-shaped translational landslides is often unexpected and covert; secondly, the field investigation and monitoring data for this kind of landslide is often absent, and the research on translational landslide lacks monitoring engineering and measured data on landslide physical parameters.

Thus, demonstrating and validating the deformation and failure mode of the translational landslide in the theoretical analysis is difficult. As shown observation point b and c in Fig. 2, the Wobaoshi landslide seriously threaten residential houses and highways, the houses had cracked and highways had uplifted, so this landslides seriously threatens the safety of people's property and transportation, thus it is necessary to research on it. This supplementary part is from line 181 to line 183. We added a photo of the house cracking in observation point b of Figure 2, and this photo was taken in a field investigation in the Wobaoshi landslide. Additionally, detail description and conclusion of characteristics and failure mode of the landslide in chapter – Introduction, have been transfered to the chapter – Results.

3. In the Landslide characteristics the geological structure is not clearly presented. Monitoring results and numerical modelling results are clearly presented (graphs, models). Modification: In origin manuscript, the Section 1 "Landslide characteristics" is not clearly presented like the comment said. Thus we reconstruct the Section 1 and especially make change on the forming conditions, especially the engineering geological conditions of this landslide. At the same time, some geological terms have been corrected. Through the modification, we feel that geological structure and characteristics of the Wobaoshi landslide will be more clearly and accurate than the origin vision.

4. Discussion is clearly written. The system of long-term gradual opening of the cracks and short-term opening and closing of cracks (blocks tilting) (very characteristic on the graphs) could be described more in detail. Modification: The system of long-term cracks opening and short-term cracks closing is described in detail in Section 2.2 "Monitoring Data Analysis", and the Section 4.1 "Deformation and Failure Mode Exploration" in discussion, is refined by the landslide monitoring data analysis in Section 2.2. And we have increased the relationship between the "Monitoring Data Analysis" and "Deformation and Failure Mode Exploration" in Section 4.1.

5. Any possible mitigation measures or additional monitoring suggested? Explanation and modification: As shown in Section 4.3, the manuscript proposes the following suitable monitoring methods referring to the plate-shaped translational landslide, such as long- period and multiple parameters monitoring, and additional monitoring suggestion is the inclination measuring device for plate girders, and this monitoring method in next working plan will be effective and suitable for this type of landslide. In origin manuscript, the optimization monitoring methods may not be very clear, so we have highlighted the new method suggested in the next step of landslide monitoring, therefore, it is beneficial to in-depth exploration of the deformation and failure mode of the landslide and it will improve the success rate of early warning.

Please also note the supplement to this comment:
https://www.nat-hazards-earth-syst-sci-discuss.net/nhess-2019-133/nhess-2019-133-AC1-supplement.pdf
* * *
[Figure]

[revised manuscript text omitted]

---

## Author Comment (AC2) · 26 Jul 2019

Dear professor DarveïijŽ We are very grateful to receive your discussion and comments about this paperïijŇand your comments and suggestions are important for improving the next research work. The research idea of this paper is to verify the theoretical deformation failure mode of the plate-shaped translational landslides based on the measured data of the key physical parameters of the landslide. As your express in your comments, we want to obtain the correlation between rainfall intensity and block motion. Next we discuss several issues in your comments. Firstly, the research target of

this paper - the Wobaoshi landslide, according to the landslide investigation and investigation report, is a typical sandstone and mudstone interbed rock landslide whose soil cover layer is very thin, and its thickness is not more than 5m from the sectional graph, as shown in Figure below(Chen et al., 2015). Therefore, when the geomechanical model is established, the cover layer is neglected, and the static geomechanical model of the plate-shaped rock sliding body is established based on limit equilibrium method. The basic characteristic of the limit equilibrium method is that the Mohr-Coulomb failure criterion of the soil in static equilibrium conditions is considered, that is, the problem's solution is solved by analyzing the destruction of the soil's balance, and this method considers soli elastic-perfectly plastic model which obey the Mohr-Coulomb failure criterion and associated flow rules. Secondly, regarding the internal friction angle of the clay, its value is measured by the geochemical experiment. On the other hand, the clay layer is severely weathered, so its internal friction angle is small. The internal friction angle $\theta$ is obtained by triaxial compression tests of the core, which is taken from the sand-mudstone contact surface in sliding surface, and the internal friction angle $\theta$ = 11.2°(Chen et al., 2015). In general, the dilatancy effect obtained by the associated flow law is much larger than the actual observation, especially in the case of lateral infinite(Tschuchnigg et al., 2015a). However, for slope stability analysis, lateral infinite is not considered in most cases, and the dilatancy effect is not significant (Griffiths & Lane, 1999). Therefore, it is reasonable to set the dilatancy angle to be equal to the internal friction angle in my paper. Additionally, the detail description finite element modeling is that, we choose MIDAS GTS NX to perform numerical simulations and calculations, and the linear elastic model finite element method is used in this paper, the elastic modulus and Poisson's ratio are shown in Table 4 in the paper. The right boundary of the model selects the position of the right side of the landslide about 30m from the foot of the slope; the lower boundary selects the elevation of 0 m; the left boundary is located inside the mountain, about 30m away from the plate girder I. The element type adopts a plane strain quadrilateral-triangle mixing element, and the whole model divides 13775 elements and 14026 nodes. The bottom boundary of the

calculation model constrains the vertical and horizontal displacement, and the left and right boundaries constrain the horizontal displacement. The model uses steady-state seepage calculation, and the water levels at the left and right boundaries were set to 342 and 275 m, respectively. The water level in the Crack I and Crack II is selected for a typical change period, as shown in the table below. The deformation mechanism of the plate-shaped landslide is studied by simulating the deformation of the plate girders caused by the change of the seepage stress field in a water level change period. The geomechanical model in this paper is relatively simple, which is a purely static method, and it ignores the influence of strain history. Moreover, in the numerical simulation calculation, the boundary conditions are set relatively simple, and the working conditions considered are also less. These shortcomings are improved in the next research. Since the interactive discussion of this paper is coming to an end, our response is in a hurry, so we are looking forward to further communication with you by Email. Thank you very much for your suggestion and consideration, and we look forward to hearing from you. Best regards, Yimin Liu, Guiyun Gao and Chenghu Wang.

ReferencesïijŽ Chen L., Liu Y., Feng X.: The investigation report on remediation project of Wobaoshi landslide, Sanhui Town, Enyang District, Bazhong City, The Institute of Exploration Technology of CAGS, Chengdu, Open File Rep., pp.57, 2015. Tschuchnigg, F., Schweiger, H.F. & Sloan, S.W. (2015a). Slope stability analysis by means of finite element limit analysis and finite element strength reduction techniques. Part I: Numerical studies considering non-associated plasticity. Computers and Geotechnics, 70, 169-177. Tschuchnigg, F., Schweiger, H.F., Sloan, S.W., Lyamin, A.V. & Raissakis, I. (2015b). Comparison of finite-element limit analysis and strength reduction techniques. Géotechnique, 65(4), 249-257. Griffiths, D.V. & Lane, P.A. (1999). Slope stability analysis by finite elements. Géotechnique, 49(3), 387-403.

[Figure]

Fig. I-I' sectional graph of the Wobaoshi landslide

**Fig. 1.**

| Loading Steps | Crack I | Crack II |
|---|---|---|
| 0 | 314.50 m | 311.00 m |
| 1 | 316.00 m | 313.00 m |
| 2 | 317.50 m | 315.00 m |
| 3 | 316.00 m | 313.00 m |
| 4 | 314.50 m | 311.00 m |

Tab. loading step of the water level in Crack I and II

**Fig. 2.**

---

## Referee Comment (RC2) · Felix Darve (Referee) · 25 Sep 2019

NHESS review report on paper 2019-133 by Y. Liu, C. Wang, G. Gao, P. Wang, Z. Hou, Q. Jiao

I have been asked first to produce a ÂńÂădiscussionÂăÂż on your very interesting paper. You will find again this discussion below, which presents frankly my point of view particularly with respect to the limits attached to the application of limit equilibrium method in geomechanics/geotechnics.

[Figure]

Now for a review report, I will insist on the interest for the reader to understand well your FEM modelling and ( if possible) to consider as instability criterion the so called ÂńÂăsecond order work criterionÂăÂż ( well described in the recent book ÂńÂăFailure in Geomaterials, a Contemporary TreatiseÂăÂż by Wan, Nicot and Darve and published by ISTE/WILEY). Your paper will become a reference paper if you are able to compare (i) limit equilibrium method, (ii) FEM computations with Mohr Coulomb criterion and (iii) FEM computations with the second order work criterion. I am not asking you to do that in the present paper – probably too much work – but envisage such comparisons for a next paper. So for the present paper, I suggest to youÂă: (i) to add in the paper some comments about the limits of the equilibrium method, (ii) to give more details about your FEM modelling. In the present state of your paper, your numerical computations appear as a black box, this is not reasonable for the readership.

Previous published discussionÂă: The Authors discuss in their paper devoted to ÂńÂăinstability conditions and failure mode of a translational landslideÂăÂż a case study, whose main interest lies on its careful long term monitoring. The characteristics of this landslide are remarkably described and these results will certainly constitute a very valuable data bank for future studies. Moreover the basic mechanical analysis of the related boundary value problem has been conducted in a very convincing manner and the failure mechanism is clearly exhibited. So all the ingredients have been collected to form a perfect basis for the modelling. Let us insist here on the fact that a numerical or analytical mechanical model of natural hazards - to be fruitful - has to take into account as properly as possible the discriminant or critical features, exhibited by preliminary geological/mechanical/pluviometric analyses. These analyses have to be based on a well designed monitoring campaign and the issueing data bank. This is clearly the case here and the Authors have to be congratulated for having achieved that – particularly the correlation between rainfall intensity and block motion.

So what is interesting to be discussed now is the choice of the modelling strategy. The analytical model is based on limit equilibrium method, whose limits have to be

recalledÂǎ: - this is a purely static method, - it ignores the influence of strain history, while it is well recognised that soils present an important hardening regime involving large plastic strains before failure, - a coulombian friction is taken into account, while the soil can fail before reaching Mohr-Coulomb plastic limit condition according to the second order work criterion, which is a more general material instability criterion. This is particularly important here, since the sliding surface has a very low slope angle of 6-8 degrees. Thus the instabilities are certainly appearing before Mohr-Coulomb criterion. - the hydro-mechanical constitutive relation of the sliding zone, which plays here a central role, can not be taken into account.

Having in mind these drastic limits, the reader is waiting for a deeper and more realistic finite element computation. However the finite element modelling is presented with very few details. According to the values of parameters as given on table 4, it seems that all involved geomaterials ( rocks and soils) have been considered as associate plastic materials. Thus their dilatancy angle is assumed to be equal to their friction angle, what is clearly not verified experimentally. Let us note that the constitutive elasto-plastic matrix is symmetric for associate materials, preventing to describe all instabilities and bifurcations occuring before the Mohr-Coulomb plastic limit surface.This is probably the reason why it has been necessary to choose a so low friction angle of $11.2°$ for clay – what seems unrealistic.

---

## Author Comment (AC3) · 26 Sep 2019

Dear professor Darve:

We are very grateful to receive your referee comments (RC) again. As you said in short comments (AC), this manuscript needs to be modified and improved in two main aspects, the first is to build the geomechanical model with more comments about the limits of the equilibrium method in Section 3.1, on the other hand is to add more details about my FEM modeling in Section 3.2. After careful discussion by author and coauthors, we found that these comments are very important for improving the quality of this manuscript.

With your help and comments, especially FEM computations with Mohr Coulomb criterion and FEM computations with the second order work criterion, we realized that the geomechanical model of landslide is relatively simple, which is a purely static method, and it ignores the influence of strain history. Moreover, in the numerical simulation calculation, the boundary conditions are set relatively simple, and the working conditions considered are also less than the actual situation. And we want to use the DEM to establish the landslide model in next research, and discuss the influence of pore water pressure on the sliding body sliding and soil damage, and compare it with the existing model. These shortcomings in this manuscript will be improved in future research.

Please see the detailed revision, and we carefully proof-read the manuscript to minimize typographical, grammatical, and bibliographical errors, and the modified parts are marked in red in the revised manuscript in supplement online.

Thank you very much for your suggestion and consideration, and we look forward to get your constructive advice.

Best regards, Yimin Liu, Guiyun Gao and Chenghu Wang.

Detailed revision

1. To add in the paper some comments about the limits of the equilibrium method.

Modification:

With the help of references about limit equilibrium method and instabilities in geomaterials, we add the following supplement in Section 3.1.

According to characteristics of the Wobaoshi landslide in Section 1.2, when the geomechanical model is established, the cover layer is neglected, and the static geomechanical model of the plate-shaped rock sliding body is established based on the limit

equilibrium method. The basic characteristic of the limit equilibrium method is that the Mohr-Coulomb failure criterion of the soil in static equilibrium conditions is considered, that is, the problem's solution is solved by analyzing the destruction of the soil's balance. And soli elastic-perfectly plastic model was chosen, which obey the Mohr-Coulomb failure criterion and associated flow rules (Darve et al., 2004; Labuz et al., 2015).

2. To give more details about your FEM modeling in the present state of your paper.

Modification:

This comment is very important for the paper, and the FEM modeling is lack of detailed description, we add the following supplement in Section 3.2.

The position of the right side of the landslide about 30m from the foot of the slope is selected as the right boundary of the model; the lower boundary is setted at the elevation of 0 m; the left boundary is located inside the mountain, about 30m away from the plate girder I. The element type adopts a plane strain quadrilateral-triangle mixing element, and the whole model is divided into 13775 elements and 14026 nodes. Here we constrain the vertical and horizontal displacement of its bottom boundary, and the left and right boundary conditions are set to constrain the horizontal displacement. The model uses steady-state seepage calculation, and the water levels at the left and right boundaries were set to 342 and 275 m, respectively. The typical pore-water-level data in the crack I and crack II presented in Table 3 were introduced into the finite element model, and were selected for a typical change period presented in Table 5.

3. The paper chose a so low friction angle of 11.2° for clay –what seems unrealistic.

Modification:

It is very interesting for the value of friction angle, and it seems too low and unrealistic, so we add the explanation and discussion in Section 4.2.

The internal friction angle, $\theta$ = 11.2°, is so low for clay, which seems unrealistic. However, the angle $\theta$ is obtained by triaxial compression tests of the core, which is taken from the sand-mudstone contact surface in sliding surface, and the internal friction angle $\theta$ = 11.2° (Chen et al., 2015). One of the reasons may be that the clay layer is severely weathered, so its internal friction angle is small. In general, the dilatancy effect obtained by the associated flow law is much larger than the actual observation, especially in the case of lateral infinite (Tschuchnigg et al., 2015a). However, for slope stability analysis, lateral infinite is not considered in most cases, and the dilatancy effect is not significant (Griffiths Lane, 1999). Therefore, it is reasonable to set the dilatancy angle to be equal to the internal friction angle.

ReferencesïijŽ

[1]Chen L., Liu Y., Feng X.: The investigation report on remediation project of Wobaoshi landslide, Sanhui Town, Enyang District, Bazhong City, The Institute of Exploration Technology of CAGS, Chengdu, Open File Rep., pp.57, 2015.

[2]Darve F., Vardoulakis I.: Degradations and Instabilities in Geomaterials, Springer Vienna, Austria, 2004.

[3]Griffiths, D. V., Lane, P. A.: Slope stability analysis by finite elements, Géotechnique, 49(3), 387-403, 1999.

[4]Labuz J. F., Zang A.: Mohr–Coulomb Failure Criterion, Rock Mechanics and Rock Engineering, (2012)45:975–979, 2012.

[5]Tschuchnigg, F., Schweiger, H.F. Sloan, et al: Slope stability analysis by means of finite element limit analysis and finite element strength reduction techniques. Part I: Numerical studies considering non-associated plasticity, Computers and Geotechnics, 70, 169-177, 2015.

[6]Tschuchnigg, F., Schweiger, H.F., Sloan, S.W., et al: Comparison of finite-element limit analysis and strength reduction techniques, Géotechnique, 65(4), 249-257, 2015.

Please also note the supplement to this comment:
https://www.nat-hazards-earth-syst-sci-discuss.net/nhess-2019-133/nhess-2019-133-AC3-supplement.pdf
* * *

[revised manuscript text omitted]

---

## Author Response (AR1)

**1. Reply to the reviewer 1**

We are very grateful to this referee comments, and we have carefully read and considered the referee's comments, and these comments are important for improving the quality of this manuscript. Based on these comments, we have made carefully modification and proofreading on the original manuscript, the detail modifications are mainly for the chapter - Abstract and Introduction, which have been reconstructed and rewrote, and description of the Wobaoshi Landslide's geological structure and characteristics have been modified and reconstructed, and language, grammar and professional terms in English also have been improved.

Please see the detailed revision, and we carefully proof-read the manuscript to minimize typographical, grammatical, and bibliographical errors, and the modified parts are marked in red in the revised manuscript.

Thank you very much for your suggestion and consideration, and we look forward to hearing from you.

Best regards,

Yimin Liu, Chenghu Wang and Pu wang.

**Detailed revision for comments of reviewer 1**

1. *Language, in general is not good and some sentences are unclear. Some professional terms are incorrect. (comment 1)*

**Modification:**

For the unclear parts and unprofessional terms in the manuscript, we have made serious changes in English expression to ensure clearly presented, and proofreading also has been done, and the revised changes are in red in the manuscript.

For instance, "overall occurrence of the rock formation" has changed to "overall attitude of rockbed" in line 131, "planar graph" has changed to "topographic map" in Fig. 3 in line 136, "gage" has changed to "gauge" in line 200, 202, 204 and 205, "long-period" has changed to "long-term", "heavy rain" has changed to "periodic rain", "rainy season" has changed to "monsoon", "mechanical" has changed to "geomechanical", "field exploration" has changed to "field investigation", "starting" has changed to "initiation", "theorical" has changed to "theoretical" in the manuscript, and we have spelled all abbreviations and acronyms at their first mention in the main text, such as enhanced thematic mapper called ETM+, and digital elevation model called DEM in line 67 and 68. Thus, these descriptions will be more accurate and professional than before.

2. *Introduction, there is a little more information (what, where, why) and presentation of the main aim of the paper. Does the landslide threaten something? Why it is important to be studied? (comment 2)*

**Modification:**

This comment is very important for the paper, and the presentation of the main aim in original manuscript is lack of refinement and detailed narrative. Therefore, we modify and supplement the purpose and importance of this manuscript, firstly, the occurrence of plate-shaped translational landslides is often unexpected and covert; secondly, the field investigation and monitoring data for this kind of landslide is often absent, and the research on translational landslide lacks monitoring engineering and measured data on landslide physical parameters. Thus, demonstrating and validating the deformation and failure mode of the translational landslide in the theoretical analysis is difficult.

As shown observation point b and c in Fig. 2, the Wobaoshi landslide seriously threaten residential houses and highways, the houses had cracked and highways had uplifted, so this landslides seriously threatens the safety of people's property and transportation, thus it is necessary to research on it. This supplementary part is from line 181 to line 183. We added a photo of the house cracking in observation point b of Figure 2, and this photo was taken in a field investigation in the Wobaoshi landslide.

[Figure]

Fig. 2 Topographic map of the Wobaoshi landslide and photographs of observation points: (a) exposed bedrock at the front edge; (b) the houses had cracked at the front edge (c) the roadbed is pushed uplifted at the front edge; (d) crack II and bent trees; and (e) crack I.

Additionally, detail description and conclusion of characteristics and failure mode of the landslide in chapter – Introduction, have been transferred to the chapter – Results.

3. *In the Landslide characteristics the geological structure is not clearly presented. Monitoring results and numerical modeling results are clearly presented (graphs, models). (comment 3)*

**Modification:**

In origin manuscript, the Section 1 "Landslide characteristics" is not clearly presented like the comment said. Thus we reconstruct the Section 1 and especially make change on the forming conditions, especially the engineering geological conditions of this landslide. At the same time, some geological terms have been corrected. Through the modification, we feel that geological structure and characteristics of the Wobaoshi landslide will be more clearly and accurate than the origin vision.

4. *Discussion is clearly written. The system of long-term gradual opening of the cracks and short-term opening and closing of cracks (blocks tilting) (very characteristic on the graphs) could be described more in detail. (comment 4)*
**Modification:**

The system of long-term cracks opening and short-term cracks closing is described in detail in Section 2.2 "Monitoring Data Analysis", and the Section 4.1 "Deformation and Failure Mode Exploration" in discussion, is refined by the landslide monitoring data analysis in Section 2.2. And we have increased the relationship between the "Monitoring Data Analysis" and "Deformation and Failure Mode Exploration" in Section 4.1.

5. *Any possible mitigation measures or additional monitoring suggested? (comment 5)*
**Explanation and modification:**

As shown in Section 4.3, the manuscript proposes the following suitable monitoring methods referring to the plate-shaped translational landslide, such as long-period and multiple parameters monitoring, and additional monitoring suggestion is the inclination measuring device for plate girders, and this monitoring method in next working plan will be effective and suitable for this type of landslide.

In origin manuscript, the optimization monitoring methods may not be very clear, so we have highlighted the new method suggested in the next step of landslide monitoring, therefore, it is beneficial to in-depth exploration of the deformation and failure mode of the landslide and it will improve the success rate of early warning.

**2. Reply to the reviewer 2# (professor Darve)**

Dear professor Darve:

We are very grateful to receive your referee comments (RC) again. As you said in short comments (AC), this manuscript needs to be modified and improved in two main aspects, the first is to build the geomechanical model with more comments about the limits of the equilibrium method in Section 3.1, on the other hand is to add more details about my FEM modeling in Section 3.2. After careful discussion by author and co-authors, we found that these comments are very important for improving the quality of this manuscript.

With your help and comments, especially FEM computations with Mohr Coulomb criterion and FEM computations with the second order work criterion, we realized that the geomechanical model of landslide is relatively simple, which is a purely static method, and it ignores the influence of strain history. Moreover, in the numerical simulation calculation, the boundary conditions are set relatively simple, and the working conditions considered are also less than the actual situation. And we want to use the DEM to establish the landslide model in next research, and discuss the influence of pore water pressure on the sliding body sliding and soil damage, and compare it with the existing model. These shortcomings in this manuscript will be improved in future research.

Please see the detailed revision, and we carefully proof-read the manuscript to minimize typographical, grammatical, and bibliographical errors, and the modified parts are marked in red in the revised manuscript in supplement online.

Thank you very much for your suggestion and consideration, and we look forward to get your constructive advice.

Best regards,
Yimin Liu, Guiyun Gao and Chenghu Wang.

**Detailed revision for comments of reviewer 2**

1. *To add in the paper some comments about the limits of the equilibrium method. (comment 1)*

**Modification:**

With the help of references about limit equilibrium method and instabilities in geomaterials, we add the following supplement in Section 3.1 from line 279 to 286.

According to characteristics of the Wobaoshi landslide in Section 1.2, when the geomechanical model is established, the cover layer is neglected, and the static geomechanical model of the plate-shaped rock sliding body is established based on the limit equilibrium method. The basic characteristic of the limit equilibrium method is that the Mohr-Coulomb failure criterion of the soil in static equilibrium conditions is considered, that is, the problem's solution is solved by analyzing the destruction of the soil's balance. And soli elastic-perfectly plastic model was chosen, which obey the Mohr-Coulomb failure criterion and associated flow rules (Darve et al., 2004; Labuz et al., 2015).

2. *To give more details about your FEM modeling in the present state of your paper. (comment 2)*

**Modification:**

This comment is very important for the paper, and the FEM modeling is lack of detailed description, we add the following supplement in Section 3.2 from line 355 to 363.

The position of the right side of the landslide about 30m from the foot of the slope is selected as the right boundary of the model; the lower boundary is setted at the elevation of 0 m; the left boundary is located inside the mountain, about 30m away from the plate girder I. The element type adopts a plane strain quadrilateral-triangle mixing element, and the whole model is divided into 13775 elements and 14026 nodes. Here we constrain the vertical and horizontal displacement of its bottom boundary, and the left and right boundary conditions are set to constrain the horizontal displacement. The model uses steady-state seepage calculation, and the water levels at the left and right boundaries were set to 342 and 275 m, respectively. The typical pore-water-level data in the crack I and crack II presented in Table 3 were introduced into the finite element model, and were selected for a typical change period presented in Table 5.

Table 5    Loading steps of the water level in Crack I and II in FEM model

| Loading Steps | Crack I | Crack II |
|---|---|---|
| 0 | 314.50 m | 311.00 m |
| 1 | 316.00 m | 313.00 m |
| 2 | 317.50 m | 315.00 m |
| 3 | 316.00 m | 313.00 m |
| 4 | 314.50 m | 311.00 m |

3. *The paper chose a so low friction angle of 11.2 ° for clay –what seems unrealistic. (comment 3)*

**Modification:**

It is very interesting for the value of friction angle, and it seems too low and unrealistic, so we add the explanation and discussion in Section 4.2 from line 445 to 453.

[revised manuscript text omitted]

---

## Author Response (AR2)

**Reply to editor and the referees**

On behalf of my co-authors, we thank editor and the referees very much for giving us an opportunity to revise our paper, we appreciate editor and reviewers very much for their positive and constructive comments and suggestions on the second version of the paper.

Based on these comments from reviewer 1#, reviewer 3# and professor Darve, we have made carefully proofreading again on language, style, grammar and professional terms in English, and the repeats in texts have been summarized and optimized, and the references cited in paper have also been corrected according to reference specifications. Combine the comments and suggestions of the three referees and editor, we focus on the supplement of detail of the geomechanical and FEM model in Section 3, and Section of discussion, especially exploration of deformation and failure mode of the landslide, and Section of conclusion have be also summarized and optimized. Please see the detailed revision of point-by-point reply, and the modified parts are marked in red in the revised manuscript.

Thank you very much for your suggestions and consideration, and we look forward to hearing from you.

Best regards,

Yimin Liu, Guiyun Gao, Pu wang, Chenghu Wang et al.

**Detailed revision for the referee's comments**

1. Line 42: Fausto G. et al., 2004; probably Guzzetti et al., 2004

**Modification:**

"Guzzetti et al., 2004" has been revised in line 42.

2. Line 43: "Research on the formation mechanism and deformation mode of a translational landslide is mainly based on two perspectives". To which research are you referring?

**Explanation:**

The two perspectives of the research of translational landslide I referred are in line 46 (Kong and Chen, 1989; Matjaž et al., 2004; Yin et al., 2005) and line 53 (Cruden et al., 1996; Emelyanova, 1986). Maybe we should add these in line 43.

3. *Line* 46: *Kong and Chen,* 1989 *is lacking in the list of references.*

**Modification:**

We are very sorry for our negligence of missing this reference, and we have added it in the list of references.

Kong J., Chen Z.: The translational landslide in red stratum located in east of Sichuan in July, 1989. Beijing: China Railway Publishing House, Landslide Column(9), 1989.

**4. Line 51-52: "...the hard rock layer covered by the upper layer (such as granite and sandstone) has a crushing effect on ...the lower weak rock layer". I don't understand. **Explanation and modification:**

What I want to express in this sentence is the upper layer contains hard rock such as granite and sandstone, and we modify it like this, "The second category includes landslides wherein the upper layer contains hard rock (such as granite and sandstone) has a crushing effect on the lower weak rock layer".

**5. Line 56-57: "Sensitive safety factors". What are? Probably you are meaning sensitivity analysis of safety factors or something like that.**

**Modification:**

As you said, "Sensitive safety factors" in Line 56-57 is the sensitivity analysis of safety factors.

**6. Line 57: Fan Xuanmei et al. (2008) is not listed in the references.**

**Explanation and modification:**

This reference cited is not by way. This reference is list in references in line 556, and it should be changed to "Fan et al. (2008)".

Fan X., Xu Q., Zhang Z., et al: Study of genetic mechanism of translational landslide, Chinese Journal of Rock Mechanics and Engineering, 27(Supp.2):3753-3759, 2008.

7. Line 62: Mario et al. (2008): I think that this reference is Floris et al., 2008 (or Floris et al., 2008). Please modify also in the reference list.

**Modification:**

We are very sorry for our negligence of missing this reference, and we have revised in the list of references and line 62.

Floris M., Bozzano F.: Evaluation of landslide reactivation: a modified rainfall threshold model based on historical records of rainfall and landslides, Geomorphology, 94(1-2): 40-57, 2008.

**8. Line 66: landslide resurrection. Are you meaning landslide reactivation?**

**Modification:**

We are very sorry for the lack of language skills, and "landslide resurrection" has been corrected to "landslide reactivation" in line 66.

**9. *Line 73: collation. I suppose data collation in the part of the text between line 43 and line 76 is not very accurate.**

**Modification:**

In the part of the text between line 43 and line 76, we elaborated the research status domestic and international of the translational landslide. We replaced "data collation" with "references collation".

10. Lines 86-87: "....laboratory physical experiments have been conducted in the to verify the failure model.". Please verify english language.

**Modification:**

The references (Xu et al. 2006; Fan et al., 2008) have been conducted laboratory physical experiments to verify the failure model, we added these references in line 87.

**11. Line 90: abtained. Probably obtained.**

**Modification:**

We are very sorry for our negligence of word spelling, and "abtained" has been corrected to "obtained" in line 90.

**12. Line 77-92: I suggest to summarise. This part of the text is too long.**

**Modification:**

We have summarized this text from line 77-92. And referee 1# encourage us to add more information (what, where, why) and presentation of the main aim of the paper, so we modify and supplement the purpose and importance of this manuscript in Line 77-92.

13. Line 100: Engineering instead Enginereing This mistake is common in the text. Please verify.

**Modification:**

We are very sorry for our negligence of word spelling, and "Engineering" has been corrected to "Engineering" in the revised vision.

14. Line 110-111: I suppose that the meaning of the sentence is "the landslide involved sandstone and mudstone belonging to the Penglaizhen Formation of the upper Jurassic". Is it right? If yes, I suggest to revise the text.

**Modification:**

You are right, we have revised in the text in line 111.

15. Lines 120-121: I suppose you are discussing about water filling primary porosity (voids between grains) and water filling secondary porosity (cracks). I suppose that the first characterises weathered rock and the second jointed rock. Is it right? If yes, I don't understand the relevance of this sentence in your description; moreover I don't understand the meaning of "trailing edge".

**Explanation and modification:**

"trailing edge" in line 121 is rear of this landslide, maybe in this sentence is not so accurate, we have deleted it in text.

**16. Lines 102-122. In my opinion the content of this part of the text doesn't deal with field survey as the title of the paragraph states.**

**Explanation and modification:**

We quite agree with your opinion. The title of this paragraph should be

"landslide location".

**17. *Line 129: 105 instead 105.**

**Modification:**

We are very sorry for our negligence of superscript spelling, and " $10^5$ " has been corrected to "105" in line 129.

18. Fig. 2: Please add some geologic data. Outcropping formation/lithology, attitude of the strata in the surveyed area. I have some doubts about your section I-I'. If the trace of the section is right, the building and the highway should be located more or less in the middle of the section? Why they are located on the SW side? The values of the contour lines are lacking in the map. I suggest to deeply revise this figure!

**Explanation and modification:**

Your comment is very constructive and significant, we had deeply revised Fig. 2, mainly added formation of lithology information in the Fig.2 based on your pertinent suggestion, and the attitude of the strata is presented in Fig 3, including argillaceous siltstone, mudstone and arkose sandstone. And you said that the position of I-I' is not so accurate, this is because we are not sure about the specific scope of the Wobaoshi landslide during the preliminary field survey, so the I-I' is not in the middle of the topographic map.

---

## Editor Decision (ED2)

**Analysis of the instability conditions and failure mode of a special type of translational landslide using long-term monitoring data: A case study of the Wobaoshi landslide (in Bazhong, China)**

Yimin Liu[a,b], Chenghu Wang[a,*], Guiyun Gao[a], Pu Wang[a], Zhengyang Hou[a], Qisong Jiao[a]

[a] Institute of Crustal Dynamics, China Earthquake Administration, Beijing, 100085, China

[b] School of Manufacturing Science & Engineering, Sichuan University, Chengdu, 611730 ,China

**Abstract:** A translational landslide comprising nearly horizontal sandstone and mudstone interbed occurred in the Ba river basin of the Qinba–Longnan mountain area. Previous studies have succeeded to some extent in investigating on the formation mechanism and failure mode of this type of rainfall-induced landslide; however, it is very difficult to demonstrate and validate the previously-established geomechanial model owing to lack of landslide monitoring data. In this study, we considered a translational landslide exhibiting an unusual morphology, ie., the Wobaoshi landslide, that occurred in Bazhong, China. First, the engineering geological conditions of this landslide were determined through field investigation, and the deformation and failure mode of the plate-shaped main body were analyzed. Second, long-term monitoring was performed to obtain multiparameter monitoring data (width of the crown crack, rainfall, and pore-water pressure). Finally, an equation was developed to obtain the critical water height of the multistage bodies, i.e., $h_{cr}$, based on the geomechanical model analysis of the multistage main bodies, and the reliability of this equation was verified using long-term monitoring data. Subsequently, the deformation and failure modes of the plate-shaped bodies were analyzed and investigated based on numerical simulations and calculations. Thus, the multiparameter monitoring data proved that the stability of the main body is majorly controlled by the rainfall and pore-water pressure, further, the pore-water pressure in the crown crack was positive with respect to the initiation of sliding of the plate-shaped bodies. Simultaneously, an optimized monitoring methodology was proposed for this type of landslide. Therefore, these research findings are theoretically and practically significant to study the translational landslides occurring in this area.

**Keywords:** Translational landslide; Long- term monitoring; Instability conditions; Failure mode; Plate-shaped main body; Pore-water pressure.

**0. Introduction**

A special type of landslide can be observed in the red beds of the Qinba–Longnan mountainous area. This landslide mainly occurres in the rock mass of the nearly horizontal sandstone and mudstone interbed located in the Ba river basin, and hasexhibits the following characteristics: The cover layer is  thin (generally not more than 5 m); the sliding surface is nearly horizontal; and the inclination angle of the rock bed is generally only $3^{°} \sim 8^{°}$. The main body of this landslide is typically a thick sandstone layer with good integrity, whereas its bottom is a weak layer comprising mudstone. During monsoon, particularly during rainstorms, the main body is pushed horizontally along the sliding surface. Some scholars have termed this phenomenon a flat-push landslide, which is a typical rainfall-induced landslide (Zhang et al., 1994; Guzzetti et al., 2004; Xu et al., 2010).

The research on the formation mechanism and deformation mode of a translational landslide is mainly classified into two categories. The first category includes the translational landslides that are primarily caused by the hydrostatic pressure or confined water pressure because of rainstorms (Kong and Chen, 1989; Matjaž et al., 2004; Yin et al., 2005). The main body of the thick sandstone can slide along the surface owing to the hybrid action of the hydrostatic pressure in crown cracks and the uplift force of the sliding surface (Wang and Zhang, 1985; Zhang et al.,

1994; Xu et al., 2006; Fan, 2007). Simultaneously, the sliding soil, which is expanded by rainwater, causes a slip between the nearly horizontal layers (Yin et al., 2005). The second category includes landslides in which the the upper layer containing hard rock (such as granite and sandstone) has a crushing effect on the lower weak rock layer, causing the lateral expansion of the rock mass, resulting in a landslide (Cruden and Varnes, 1996; Emelyanova, 1986).

With respect to the theoretical study on rainfall-induced translational landslide, scholars and researchers worldwide have used physical simulation experiments, gemechanical model analysis, and satellite remote-sensing methods to investigate the genetic mechanism, initiation criteria, and sensitivity analysis of the safety factors. Fan et al. (2008) reproduced the deformation and failure process of the landslides via a physical simulation, and further verified the formation mechanism as well as the initiation criterion formula of the flat-push landslide previously studied by Zhang et al. (1994). Sergio et al. (2006) investigated the influence of the pore-water pressure on the stability of the rainfall-induced landslides, as well as the soil failure mode based on pore-water pressure via simulation experiments. Floris and Bozzano (2008) and Teixeira et al. (2015) obtained rainfall data based on the historical periodic rainfall conditions, and used physical experiments to establish an optimization model for rainfall-induced landslide initiation criteria for landslides in the southern Apennines and shallow landslides in northern Portugal; they also evaluated the landslide susceptibility and safety factors to evaluate the possibility of landslide reactivation induced by rainstorms. Barlow et al. (2003), and Martin and Franklin (2005) used the US land satellite called enhanced thematic mapper (ETM+) and the digital elevation model (DEM) data to detect the residues of translational bedrock landslides in an alpine terrain. Bellanova et al. (2018) used resistivity imaging to investigate the Montaguto translational landslide that occurred in the southern part of the Apennines; they also established a refined geometric model to observe the lithologic boundaries, structural features, and lateral and longitudinal discontinuities associated with the sliding surfaces.

Scholars have conducted further research on the formation characteristics and genetic mechanism of translational landslides by analyzing the previous studies that have collated and analyzed the current research status of translational landslides. Based on the results of the previous conducted studies, this study mainly focuses on the following two aspects.

(1) The occurrence of plate-shaped translational landslides is often unexpected and covert.

The plate-shaped translational landslides are primarily induced by rainfall; such events often occur in the red-bed zone of the Qinba–Longnan mountainous area. Because of the dense population and infrastructure observed in this area, the plate-shaped landslides, characterized by large volumes of mass, and covert and abrupt occurrence, often cause massive property loss and casualties. During the previous field investigation, such destructive events are often classified as small-scale bedrcok collapses, instead of focusing on the hidden dangers associated with landslides.

(2) The field investigation and monitoring data for this type of landslide are often unavailable.

In previously conducted studies, specific geomechanical and physical models have been established based on the historical rainfall records, and physical experiments have been conducted in the laboratory to verify the failure model (Xu et al. 2006; Fan et al., 2008). However, long-term on-site monitoring data and related analysis except the remote observation based on synthetic aperture radar (SAR) or satellite for such landslides, have not been presented in previous studies.

Therefore, several key field monitoring parameters, including the width of the crown crack, real-time rainfall, pore-water pressure, and groundwater level, should be evaluated to investigate and validate the deformation as well as failure mode of the translational landslides, and utilized to establish a geomechanical model.

Based on the formation mechanism of the translational landslide that has been established previously, we selected a typical and specific translational landslide (the Wobaoshi landslide) occurring in the Ba river basin of the Qinba–Longnan mountainous area, and conducted field investigation, long-term monitoring (February 2015 to July 2018), geomechanical model analysis, and numerical simulation to investigate the instability conditions and variation failure modes of this translational landslide under the influence of periodic rainfalls.

**1.  Characteristics of the Wobaoshi Landslide**

**1.1. Landslide Location**

The Wobaoshi landslide is located in the Ba river basin in the Qinba–Longnan mountainous area. Its specific location is in Baiyanwan village, Sanhui town, Enyang district in Bazhong, China. Fig. 1 presents the geographical location and elevation information. The Wobaoshi landslide occurred on the left bank of the Shilong river. The front edge of this landslide is in the curved section of the river, whereas its left boundary gully was observed on the concave bank on the river's left bank. The landslide area is classified as a red-bed layer in the low mountainous area, the vegetation of its main body is dense, and its geomorphic unit is cuesta structural slope. The geologic structure of the body lies toward the south side of the Nanyangchang anticline, and the landslide involved sandstone and mudstone belonging to the Penglaizhen Formation of the upper

Jurassic series (Chen et al., 2015).

[Figure]

Fig. 1 Geographic location and elevation map of the Wobaoshi landslide.

This landslide occurred in the eastern subtropical monsoon climate region, where rainfall is abundant and mostly concentrated between May and October, accounting for 75% - 85% of the total annual rainfall. The monthly average rainfall is greater than 100 mm, the largest amount of rainfall occurs in July, which has a monthly average rainfall of more than 200 mm. Further, rainstorms can be frequently observed during July, and the rainfall in this region gradually decreases after August (Chen et al., 2015). The types of groundwater are mainly fissure water in weathered bedrock and pore-water in the cracks, and the dynamic change of groundwater is considerably affected by climatic change.

**1.2. Landslide Characteristics  Conditions**

The main body has a flat long rectangular shape on the plane according to the satellite remote sensing data and landslide survey. Its longitudinal (sliding) direction is nearly 32 m, lateral length is 160 m, the average thickness of the sliding body is approximately 30 m, and its volume is approximately $1.536 \times 10^5$ m$^3$ (Chen et al., 2015). This main body belongs to small- to medium-sized landslides according to the typical scale (Ministry of Land and Resources of the

PRC; 2006). The sliding direction of the landslide is 249°, and the inclination degree of the rockbed is 6° ~ 8°. The strike in this landslide is almost parallel to the overall trend of the bank slope, which is a gently inclined bedding rock landslide. Fig. 2 presents the schematic map of the Wobaoshi landslide and photographs of five observation points. Fig. 3 presents the I-I' cross section graph of the landslide.

[Figure]

Fig. 2 The schematic map of the Wobaoshi landslide and photographs of the observation points: (a) exposed bedrock at the front edge; (b) the houses had cracked at the front edge (c) the roadbed is uplifted at the front edge; (d) crack II and bent trees; and (e) crack I.

[Figure]

Fig. 3 The I-I' cross section graph of the landslide

As shown in Fig. 2, the landslide is in a flat shape integrally, and its lengthwise is considerably smaller than the crosswise on the plane, and even smaller than the thickness of the main body. Therefore, this body can be easily mistaken for a bedrock collapse during investigation of geological hazard. According to Fig. 3, the main bodies are almost perpendicular to the potential sliding surface, and a group of long and straight structural planes located parallel to the slope cut the slope into two narrow plates (bodies I and II); furthermore, the  surface of the main body contain cracks, both sides of the crack are closed, and the bottom of the crack is filled with clay, gravel and collapse debris (Chen et al., 2015).

The main body of the Wobaoshi landslide resulted in the formation of two main cracks from the outside to the inside, which cut and disintegrated the main body into plate-shaped blocks from front to back, as shown in the photographs of the observation points c and d in Fig. 2. The plate-shaped bodies I and II are aslo presented in Fig. 2. The landslide is a two-stage translational landslide in which the longitudinal length of body I is 12 m, the identifiable lateral width and thickness of which are approximately 70 and 30 m, respectively, and the longitudinal length of body II is 16 m, the identifiable lateral width and thickness of which are approximately 65 and 28 m, respectively. Body I forms crack I with the crown of the landslide, whereas body II forms crack II with body I. When large-intensity rainfall occurs during monsoon, pore-water can be observed in the cracks, indicating that cracks I and II exhibit preferable water-storage conditions.

As denoted by the photograph of the observation point d in Fig. 2 shows, bent trees grow on the crown of the landslide bodies I and II. The trees on the landslide are skewed with the sliding of the soil mass, after the sliding stops, the upper part of the trunk becomes more upright with each passing year. The existence of bent trees represents the tendency of the slope body to become unstable or that the existing landslide accumulation body may slide again, this is also historical evidence of the slow sliding movement of landslides (Zhang et al., 2015).

 the shallow surface of the Wobaoshi landslide is a 2–3 m thick layer comprising collapsed and plowed soil. The main body contains  thick sandstone with good integrity, whereas the bottom sliding surface is a weak interlayer  of silty mudstone. Thus, the Wobaoshi landslide is a typical and special translational landslide, it can be considered to be a plate-shaped landslide  (Fan et al., 2008; Xu and Zeng, 2009; Xu et al., 2010).

The engineering geologic conditions of the Wobaoshi landslide can be observed based on its characteristics, i.e., the rapid immersion of groundwater softens the joint surface of soil and rock formation, especially under a rainstorm. Then, the group of open cracks located parallel to the slope in the main body is concentrated and quickly filled with water; subsequently, the groundwater level rises and the pore-water pressure increases drastically, such that bodies I and II will slide along the contact surface of the bottom sand-mud-rock weak layer. This condition changes the stress mode and equilibrium state of the rock and soil mass, easily inducing a landslide. As can be observed at the observation point b and c in Fig. 2, the Wobaoshi landslide pose a major threat to residential houses and highways, the houses cracked, and the highways were uplifted on its front edge, therefore, this landslide considerably threatens the safety of people's property and transportation.

**2. Landslide Monitoring Scheme and Monitoring Data Analysis**

**2.1. Long-term Monitoring Scheme**

According to the detailed investigation of the Wobaoshi landslide, two cracks are observed to extend through the sliding surface at the crown of the landslide. As the hydrostatic pressure in the cracks strongly influences the stability of the plate-shaped landslide (Fan et al., 2008; Guo et al., 2013), via real-time monitoring of cracks, rainfall and pore-water pressure measurements were conducted from February 2015 to July 2018 to determine the state of landslide during different periods such as rainy and non-rainy seasons, together with the interaction between multistage main bodies and the sliding surface. Fig. 4 shows the layout graph of the monitoring equipment.

[Figure]

Fig. 4 Layout planar graph of the monitoring equipment

As shown in Fig. 4, two non-contact crack automatic monitors, LF I and II, are installed on both sides of cracks I and II, respectively, to record the real-time variation of the width with respect to the two cracks (Liu et al., 2015). An automatic rain gauge is installed in a flat space and no tree occlusion is observed at the crown of the landslide to record real-time and cumulative rainfall values. Two pore-water pressure gauges are installed at the bottom of cracks I and II to measure the pore-water pressure. The pore-water level, $h_c$, can be calculated using $h_c = H - h_i + h_m$, where $hi$ is the installation depth of the pore-water pressure gauge, $H$ is the depth of the crack, and $h_m$ is the measured the pore-water pressure gauge.

In this example, the initial width of crack I is 5.640 m, whereas the initial width of crack II is 4.492 m (the first measurement was conducted in January 2015); the installation depth $h_{i1} = 24.72$ m, and the depth of crack I is $H_1 = 38$ m, with $h_{c1} = 13.28m + h_{m1}$. Additionally, the installation depth $h_{i2} = 24.85$ m, the depth of crack I is $H_2 = 35$ m, and $h_{c2} = 10.15m + h_{m2}$. The monitoring frequency of the crack width is thrice a day, the monitoring frequency of the pore-water pressure is twice a day, and the accumulative value for one month is considered to be the amount of rainfall. The multiparameter monitoring data are transmitted to the monitoring server through the GPRS network.

[Figure]

 (a)           (b)           (c)
Fig. 5 Installation of the monitoring instrument: (a) crack I gauge; (b) rain gauge and pore-water pressure gauge; (c)
crack II gauge.
## 2.2. Monitoring Data Analysis

the monitoring work conducted with respect to the

Wobaoshi landslide for three-and-a-half years (February 2015 to July 2018); the details of these monitoring data are in presented in Tables 1 and 2. The  time curves in Fig. 6 denote the monitoring data with respect to the amount of rainfall and the width of cracks I and II. Figs.

7(a) and 7(b) present the comparison curves of the monitoring data based on the width of cracks I

and II with respect to their pore-water pressures, respectively.

[Figure]

Fig. 6 The monitoring data curves (amount of rainfall and width of cracks I and II)

[Figure]

       (a)                     (b)

Fig. 7 The monitoring data curves: (a) width of crack I and its pore-water pressure; (b) width of crack II and its
pore-water pressure.

Based on the  comparison and analysis of the data curves in Figs. 6 and 7, the

Wobaoshi landslide is still considered to be in the creep deformation state, and the plate-shaped body exhibits a regular trend with respect to the changes in amount of rainfall and pore-water pressure. Cracks I and II show a preferable water-storage capacity during monsoon, and the crack width variation is affected by the increasing pore-water pressure. The specific analysis can be given as follows.

(1) A clear correspondence can be observed between the absolute extension value of the crack width and season variation (i.e.  amount of rainfall); the  amount of rainfall can be used to determine the variation of the width of the two cracks. As shown in Fig. 6, the widths of cracks I and II increase with an increase in the amount of rainfall during monsoon (May to September), whereas their crack widths gradually decrease during the non-rainy seasons (October to April ). As indicated in Fig. 8, the maximum width of crack I is 6.615 m, and its absolute extension value is approximately 1 m during July ~ August in 2017 (during which the monthly rainfall is greater than 250 mm). The maximum width of crack II is 5.40 ~ 5.45 m, and its absolute extension value is also approximately 1 m during July ~ August in 2015 and 2017. During the non-rainy seasons, when the amount of rainfall decreases, the crack width decreases and the minimum value can be observed in January of each monitored year.

[Figure]

Fig. 8 The curves of the absolute extension value of crack I and II

(2) The width of cracks I and II tend to increase year by year, indicating that the two-stage body of the Wobaoshi landslide is still moving owing to the influence of rainfall. In Fig. 6, the data obtained during the monitoring period indicate that the minimum widths of crack I and II

gradually increase and that, their  value is considerably affected by the amount of rainfall .

(3) Fig. 7 shows that the stretching of cracks I and II, or of both the curves follows the same trend as the pore-water pressure, i.e., the magnitude of pore-water pressure determines the width variation of the cracks. Fig. 7(a) shows that crack I exhibits good water-storage during monsoon, after the main body slides, a certain pore-water level can be maintained because of rainfall.

Additionally, the increase in amount of rainfall increases the water level in the cracks, and the increase in pore-water pressure positively affects the initiation of the main bodies. The curves in

Fig. 7 denote that the increase in pore-water pressure has a significant causal relation with the stretching of the cracks.

**3. Model Calculation and Numerical Simulation**

A geomechanical model of plate-shaped bodies was established to obtain a generic model with respect to the evolution process in case of the Wobaoshi landslide; further, its stability was estimated, and it was used along with the monitoring data for performing comparative analysis.

According to previous studies (Fan, 2007; Fan et al., 2008; Xu et al., 2010), the water pressure on both sides of each plate-shaped body attains a balanced state, except for the outermost body, when many penetrating cracks are located parallel to the slope in the rock mass and filled with water.

However, once the outer body slides, the surrounding water pressure immediately following the plate-shaped body becomes unbalanced and new sliding damage is induced owing to the sudden decrease in the pore water level in the crown crack (Fan, 2007; Xu et al., 2010).

**3.1. Model Establishment and Stability Calculation**

According to the characteristics of the Wobaoshi landslide presented in Section 1.2, the cover layer is neglected when establishing the geomechanical model, and a static geomechanical model of the plate-shaped rock body is established on using the limit equilibrium method. The basic characteristic of the limit equilibrium method is that the Mohr-Coulomb failure criterion of the soil under static equilibrium conditions is considered, i.e., the problem can be solved by analyzing the destruction of the soil's balance (Vardoulakis, 1983). Further, the soli elastic-plastic ideal model, which obeys the Mohr-Coulomb failure criterion and the associated flow rules, is selected (Darve and Vardoulakis, 2004; Labuz and Zang, 2015).

In this study, we selected a typical section of plate-shaped bodies and established the geomechanical model, as shown in Fig. 9, with respect to the failure mode of the two-stage body of the Wobaoshi landslide. In this section, the outer layer body II is subjected to stability analysis; subsequently, the inner body I is analyzed.

[Figure]

      Fig. 9 Geomechanical model of the two-stage plate-shaped bodies

In Fig. 9, $\alpha$ denotes the angle of the sliding surface, $h_1$ and $h_2$ are the heights of the pore-water levels in cracks I and II, respectively, $L_1$ and $L_2$ are the widths of bodies I and II, respectively, $L_{c2}$ is the distance between bodies I and II; $H_1$ and $H_2$ are the heights of bodies I and

II, respectively, and $W_1$ and $W_2$ are the self-weights of body I and II per unit width, respectively.

According to the relation between the stability coefficient of the main body, $K$, and the height of the pore-water level, $h$, shown in Fig. 7 (Zhang et al., 1994; Xu et al., 2010), the stability coefficient, $K_2$ of the outer layer body II can be obtained as follows when considering the internal cohesive force of the sliding surface.

$$K_2 = \frac{\left( W_2 \cos\alpha - \frac{1}{2}\gamma_w h_2 L_2 - \frac{1}{2}\gamma_w h_2^2 \sin\alpha \right)\tan\theta + cL_2}{\frac{1}{2}\gamma_w h_2^2 \cos\alpha + W_2 \sin\alpha}. \tag{1}$$

Here, $c$ is the internal cohesion of the sliding surface; $\gamma_r$ is the unit weight of the saturated volume of sandstone; $\gamma_w$ is the unit weight of water; and $W = H{\cdot}L{\cdot}\gamma_r$. $K_2$ is set to 1, i.e., body II is set in a critical sliding state (GB/T 32864-2016, 2017). Eq. (2) is derived from Eq. (1) and can be used to calculate the maximum pore water level of body II $h_{cr2}$.

$$h_{cr2} \approx \frac{1}{2\cos\alpha}\left[ L_2^2 \tan^2\theta + \frac{8}{\gamma_w}\left( W_2 \cos\alpha \tan\theta - W_2 \sin\alpha + cL_2 \right)\cos\alpha \right]^{\frac{1}{2}} \tag{2}$$
$$- \frac{L_2}{2\cos\alpha}\tan\theta$$

According to the experimental data obtained based on the triaxial test of the Wobaoshi landslide's rock core (Chen et al., 2015), the internal friction angle of the sliding surface is $\theta = 11.2°$, the unit weigth of the saturated volume of the sandstone is $\gamma_r = 19.2$ kN/m$^3$, the unit weight of clear water is $\gamma_w = 9.8$ kN/m$^3$, and the internal cohesion of the sliding surface is $c = 10.2$ kPa. According to the sectional graph of the Wobaoshi landslide (Fig. 2), $H = 35$ m, $L = 16$ m, and $\alpha = 6°$. Therefore, according to Eq. (2), $h_{cr2} = 13.896$ m.

Based on the stability analysis of body II, using Eq. (1) and (2), and Fig. 7, the stability coefficient $K_1$ of the inner layer body I can be obtained using Eq. (3). In addition, $h_2' = h_2 - L_{c2}$ $sin\alpha$ and $L_{c2} = 3.8$ m; therefore, $h_2' = 13.499$ m.

$$K_1 = \frac{\left[ W_1 \cos\alpha - \frac{1}{2}\gamma_w \left( h_1 + h_2' \right) L_1 - \frac{1}{2}\gamma_w \left( h_1^2 - h_2'^2 \right)\sin\alpha \right]\tan\theta + cL_1}{\frac{1}{2}\gamma_w \left( h_1^2 - h_2'^2 \right)\cos\alpha + W_1 \sin\alpha} \tag{3}$$

Similarly, $K_1$ is set to 1; in body I, $H_1 = 38$m, $L_1 = 12$m, $\alpha = 6°$, $h_2' = 13.499$m, therefore, the maximum pore-water level $h_{cr1}$ of body I can be calculated using the Eq. (3) and $h_{cr1}$ = 17.249m.

The preceding calculation results show that the pore water pressure triggers the plate-shaped bodies when the pore water level at the rear of the body reaches the maximum height at which the landslide begins, i.e., when $h_{cr1}$ = 17.249 m and $h_{cr2}$ = 13.896 m. In the next section, the pore water monitoring data, acquired via landslide monitoring, are used to verify Eq. (2) and (3).

The pore-water monitoring data presented in Section 2.2, acquired via long-term monitoring, were used to verify the equation applied to calculate the maximum height of the multistage bodies,

$h_{\overline{cr}}$. According to the monitoring data obtained with respect to the pore-water pressure and installation depth of the sensors, the actual calculated maximum height values $h_{c1}$ and $h_{c2}$ of the pore-water level are presented in Table 3. Combined with the change in the absolute extension value in Fig. 8, the typical data with respect to the measured pore-water level are selected, corresponding to a sudden increase in absolute slippage (see Table 3 for details), as shown in Fig.

10.

[Figure]

(a) Crack I                (b) Crack II

Fig. 10 Determination of the maximum measured pore-water level $h_{cr}^{'}$

The dotted boxes in Fig. 10 denote the pore-water level when the bodies are sliding or tilting, i.e., the maximum pore-water level, $h_{cr}^{'}$, which causes the main body to be unstable. The measured

$h_{cr}^{'}$ in Fig. 10 can be compared with the relation between the pore-water level, $h$, and the stability coefficient of the body, $K$, obtained using Eqs. (1) and (3), respectively, which are also depicted in

Fig. 11.

[Figure]

      (a) Crack I                    (b) Crack II

     Fig. 11 Comparison of $h_{cr}^{'}$(measured) and $h_{cr}$ (theoretical)

In Fig. 11, the curves of the h-k relation represent Eqs. (1) and (3). The values of $h_{cr}^{'}$

(measured) in Fig. 12 denote that majority of the monitoring pore-water levels are not higher than the theoretically calculated levels. The Wobaoshi landslide monitoring example shows that in majority of cases, the pore water pressure causes the main body to become unstable when $h_{cr}^{'}$

(measured) $\leqslant$ $h_{cr}$ (theoretical).

**3.2. Numerical Simulation of the Plate-shaped Main Bodies**

Numerical simulations and calculations were performed with respect to the main bodies using the MIDAS GTS NX geotechnical finite element software. First, the 1:1 main body model presented in Fig. 9 was introduced into the aforementioned software, and the mechanical parameters of the main body model, i.e., elastic modulus, Poisson's ratio, gravity internal cohesion and friction angle, were defined as shown in Table 4. The left and right boundaries were located at a distance of approximately 30 m from bodies I and II respectively, and the lower boundary is observed to be located at sea level. A plane strain quadrilateral–triangle mixing element is considered, and the entire model is divided into 13775 elements and 14026 nodes. Here, we constrain the vertical and horizontal displacement of its bottom boundary, and the left and right boundary conditions are established to constrain the horizontal displacement. The model uses steady-state seepage calculation, and the water levels at the left and right boundaries were 342 and 275 m, respectively. The boundary conditions were set as follows.

(1) In case of the displacement boundary, the left and right boundaries constrained the displacement in the X-direction; i.e., TX = 0. In case of the bottom boundary, the displacements in the X and Y directions were constrained; i.e., TX = TY = 0.

(2) In case of the seepage conditions, the water levels at the left and right boundaries were set to 342 and 275 m, respectively.

The typical pore-water-level data of the four cycles obtained from 2015 to 2018 with respect to crack I and II (presented in Table 3 and Fig. 10) were introduced into the finite element model, and selected for a typical cycle change period presented in Table 5, followed by numerical calculations to obtain the typical deformation and displacement states of the plate-shaped bodies during the rainy and non-rainy seasons, as shown in Fig. 12.

[Figure]

(a) The initial state  (b) Tilt and slide occurs with an increase in pore-water level

[Figure]

(c) The bodies slide to the maximum state     (d) Bodies tilt backward when the water level decreases

Fig. 12 Example of finite element simulation and numerical calculation

The initial displacement state in Fig. 12(a) is set to zero for performing the following analysis. Fig 12(b) shows that, the multistage bodies deform horizontally along the sliding surface under the combined effect of pore water pressure and seepage. In Fig. 12(c), the multistage bodies have slid to the maximum distance, where the maximum distance of slider II is 0.945 m, which is approximately similar the value obtained in the monitoring data. In Fig. 12(d), bodies I and II exhibit the same tendency of tilting backward owing to the decrease in pore water level during the non-rainy season. Therefore, the calculation results obtained via the numerical simulations can corroborate the main body mechanics model and landslide monitoring data.

**4. Discussion**

The deformation or sliding movement of the nearly horizontal bedrock slope is almost impossible according to the traditional theory of granular equilibrium limit, and the likelihood of occurrence of a landslide is minimal. However, the special structure of translational landslide develops mainly in the Qinba–Longnan mountainous area when investigating geological hazards. Therefore, the characteristics of the plate-shaped landslide and the deformation and failure modes should be combined during the investigation and risk assessment of geological hazards to detect the hidden dangers associated with the geological conditions of the landslide. Based on the results obtained in previous studies and the monitoring results, the discussion presented in this study can be divided into the following three parts.

**4.1. Deformation and Failure Mode Analysis of the Wobaoshi Landslide**

The monitoring results of the Wobaoshi landslide can be used to validate the rainfall-triggered failure mode of the translational landslide (Zhang et al., 1994). Based on the landslide monitoring data, particularly the change trend with respect to the opening and closing of the cracks presented in Section 2.2, and the numerical simulations of the plate-shaped bodies presented in Section 3.2, the deformation and failure modes were obtained with respect to the landslide, as shown in Fig. 13. Fig. 13 shows the deformation of the plate-shaped bodies of the Wobaoshi landslide during the monitoring period (non-rainy, season–rainy and season–non-rainy season). As shown in Fig. 13(b), the large amount of rainfall during monsoon causes the infiltration of cracks with water; when the pore-water level reaches the maximum height at which the landslide begins, the increased pore-water pressure positively affects the initiation of the plate-shaped body (Fan et al., 2007). The plate-shaped landslide can be triggered when the pore water pressure increases to the threshold value. In the monitoring case, the pore-water pressure can push the plate-shaped body by approximately 1 m, resulting in the uplift of residential houses and highways on its leading edge. Therefore, one or more penetrating cracks are likely to be parallel to the slope in the body. With the approach of the rainy season, the plate-shaped body II begins to slide firstly and the water-pressure balance in cracks is destabilized. This condition causes the gliding of the plate-shaped body I, forming a multistage translational landslide via characteristic step-by-step backward movements.

[Figure]

(a) Initial State of Main bodies    (b) Sliding in Rainy Season    (c) Tilting in Dry Season

Fig. 14 Schematic of the deformation and failure mode of the Wobaoshi landslide

As shown in Fig. 13(c), the body is tilted to the crown of landslide because of the lower pore-water level and its own weight when there is less rainfall during the non-rainy season, causing the body to fall backward (inside the slope) until the top of the body is in contact with the slope surface, the crack width begins shrinking, and a narrow A-shaped crack is formed. The monitoring data of the Wobaoshi landslide and numerical simulation of the plate-shaped body can be used to verify the deformation and failure mode of the plate-shaped landslide after its occurrence (Xu et al., 2010). With each passing year, the cracks at the bottom of the slab-shaped body increase in size, and the degree of inclination of the body also continues to increase. The degree of arching of the front edge also increases, causing the stability of the landslide to decrease continuously, posing a high risk for the houses and roads located toward the front edge of the landslide.

**4.2. Determination of the Maximum Pore-water Level $h_{cr}$**

The theoretical analysis and stability calculation of the geomechanical model of the body is described in Section 4.1, along with those of the initiation criterion for multistage main bodies in case of translational landslides, i.e., determination of the maximum water height in the crack, $h_{cr}$, (Zhang et al., 1994) and calculation of the body's stability coefficient, $K$, (Xu et al., 2010), which can be determined by theoretically calculating the strata inclination, shape, weight, and physical properties (unit weigth of the saturated volume, $\gamma_r$, internal cohesion of the sliding surface, $c$, and internal friction angle of the sliding surface, $\theta$) based on the limit equilibrium theory (Lin et al., 2010). Therefore, the stability coefficient of the landslide is observed to exponentially decrease with an increase in the water-filling height of the crown crack (Fan, 2008; Xu et al., 2010).

The internal friction angle, $\theta = 11.2°$, is considerably low for clay, and seems unrealistic. However, the angle $\theta$ is obtained via triaxial compression tests of the core, obtained from the sand-mudstone contact surface in the sliding surface, and the internal friction angle $\theta = 11.2°$ (Chen et al., 2015). This may be because the clay layer is severely weathered, resulting in a considerably small internal friction angle. Generally, the dilatancy effect obtained via the associated flow law is considerably larger than the actual observation, especially in the case of lateral confinement (Tschuchnigg et al., 2015a). However, in case of slope stability analysis, lateral infinite is mostly not considered, and the dilatancy effect is not significant (Griffiths &

Lane, 1999). Therefore, it is reasonable to set the dilatancy angle equal to the internal friction angle.

In this case, with respect to the equation for calculating the maximum pore-water level, $h_{cr}$, deduced in Section 3.1, we can observe that the measured maximum pore-water level, $h_{cr}^{'}$, is close to the theoretical maximum pore-water level, $h_{cr}$, by comparing with the measured data obtained in case of the Wobaoshi landslide in Section 2.2, validating the calculation equation of $h_{cr}$, and the instability conditions of the main bodies. Additionally, the measured data in Table 3 are slightly smaller than the theoretical calculation value, i.e., $h_{cr}^{'} \leq h_{cr}$. Thus, when compared with the equation to calculate the maximum water height proposed by Zhang et al. (1994) and the physical simulation experiment conducted by Fan et al. (2008), the monitoring case of the Wobaoshi landslide shows that the $h_{cr}^{'}$ with respect to the measured data is mostly lower than the theoretical calculated value, $h_{cr}$, which can destabilize the main body. This instability may be attributed to the fact that the actual cohesion value $c'$ of the sand-shale contact surface is smaller than the cohesive force value $c$ of the sliding surface in Eq. (2) during the creep state of the landslide for a long duration or that the frictional angle of the sliding surface, $\theta$, changes slightly. According to the calculation of the stability coefficient, $K$, in Eq. (2), when $c' \leq c$, $h_{cr}^{'} \leq h_{cr}$, the body slides when $h_{cr}^{'}$ (measured) is not larger than $h_{cr}$ (theoretical).

**4.3. Optimization Methods of Landslide Monitoring**

In this study, we propose a long-term monitoring method containing more parameters based on the characteristics of the plate-shaped translational landslides by focusing on them in accordance with the existing field-monitoring-result experience as well as deformation and failure mode exploration.

First, long- term monitoring should be conducted to obtain sufficient monitoring data,  including  the groundwater level, pore-water pressure, amount of rainfall, and displacement data on the front edge of the landslide during monsoon, as well as focusing on the change of the overall inclination of the body during the non-rainy season. This is because the inclination angle α relative to the sliding surface also changes after the body slides. Thus, an inclination measuring device, which comprises a three-axis accelerometer and electronic compass should be installed in the main body to verify the theoretical exploration of the deformation mode of the plate-shaped body during the non-rainy season in Fig. 13(c). Furthermore, a sensitivity analysis of the various parameters affecting the stability coefficient K of the main body (including the pore-water level, internal cohesive force in saturated water, internal friction angle of the sliding surface, and inclination angle of the body), should be conducted based on the monitoring data. Therefore, a detailed analysis and exploration of the deformation and failure mode of the plate-shaped landslide would be beneficial and improve the success rate of landslide warning.

**5.  Conclusions**

By considering Wobaoshi landslide as an example, we use research methods, including field exploration, long-term monitoring engineering, geomechanical model analysis, and numerical simulation, to analyze the instability conditions and failure characteristics of a special type of translational landslide. The obtained research results are beneficial with respect to the stability analysis and evaluation of this type of landslide. Targeted monitoring methods are proposed to enrich theoretical research on translational landslides. The following conclusions can be obtained.

(1) The characteristics, formation conditions, and occurrence mechanism of the rainfall-triggered plate-shaped landslides are summarized in this study. Such landslides can be generally observed in a consequent slope where the inclination angle of the sliding surface is observed to be less than 10°, and a group of long and straight structural planes observed parallel to the slope cuts the slope into several narrow plates. The plate-shaped body generally contains extremely thick sandstone, which is nearly horizontal and exhibits good integrity. The bottom sliding zone is a weak mudstone interlayer that is affected by the periodic rainfalls. In addition, single or multistage plate-shaped bodies slide horizontally along the bottom mudstone sliding zone.

(2) The relation between the stability coefficient of the multistage body $K$ and the pore water level $h$ was obtained based on the mechanical model of the plate-shaped bodies, and the maximum pore water level $h_{cr}$, which causes the instability of the multistage bodies, was calculated. The instability conditions of the plate-shaped bodies were also determined.

(3) The theoretical conclusions of the plate-shaped landslide research were verified based on the long-term monitoring data. The multiparameter monitoring data denote that the stability of the main body is considerably affected by the rainfall intensity and pore water pressure. The pore water pressure in the crack is positive at the beginning of the plate-shaped body, demonstrating the rainfall-triggered failure mode of the translational landslide. In this study, we compare and analyze the measured maximum pore water level $h_{cr}^{'}$ as well as the theoretical calculated value $h_{cr}$ and discuss the influence of the variation of the internal cohesive force and internal friction angle on the stability coefficient of the main body.

(4) Based on landslide numerical simulation, we analyze and explore the deformation and failure modes of the plate-shaped landslide, i.e., the main bodies are considered to slide horizontally along the contact surface of the bottom sand–mud–rock weak layer based on the pore water pressure in the crack and the seepage effect during monsoon. During the non-rainy season, the pore water pressure decreases and disappears; the main body will be inclined to the crown of the landslide owing to its dead weight. Thus, in this study, we propose 
[revised manuscript text omitted]

---

## Author Response (AR3)

**Reply to editor**

Dear editor,

Many thanks for your great comments, and we thank you for giving us an opportunity to revise this manuscript. The manuscript had been edited and revised by English-speaking editing agency (enago), and the references cited are also by way. Some terminologies about landslide such as "main body", "crown", "crown crack", "sliding surface" et al. have been corrected in the paper (Highland and Bobrowsky, 2013). We had revised "sliding body" to "main body", "plate girder" to "plate-shaped body", "trailing edge" to "crown" in this revised manuscript and figures. And the Fig. 2 and 3 have been revised according to your comments, and we have enlarged the font dimension both in Figs. 10, 11 and 12.

Please see the detailed revision for point-by-point reply, and the modified parts are marked in red in the revised manuscript.

Thank you very much for your suggestions and consideration, we believe that the quality of the paper has been greatly improved after this revision, and we look forward to hearing from you.

Best regards,

Yimin Liu, Guiyun Gao, Pu wang, Chenghu Wang et al.

Lynn Highland, Peter Bobrowsky. The Landslide Handbook—a Guide to Understanding Landslides: A Landmark Publication for Landslide Education and Preparedness[M]. Springer Berlin Heidelberg, 2013.

[Figure]

**Detailed revision for editor's comments**

1. *Page 1*

**Explanation and modification:**

"special type" in line 2 means that the shape of the Wobaoshi landslide is special, especially the plate-shaped main body and its sliding mode, and the paper focus on special type of the deformation and failure mechanism of the landslide, so we think "special type" in title is reasonable.

We have revised in line 10 and line 18 to avoid ambiguity.

According to *The Landslide Handbook* (Highland and Bobrowsky, 2013), "sliding body" in line 17 and 19 had been corrected to "main body", and "sliding body" in the manuscript had been revised.

Lynn Highland, Peter Bobrowsky. The Landslide Handbook—a Guide to Understanding Landslides: A Landmark Publication for Landslide Education and Preparedness[M]. Springer Berlin Heidelberg, 2013.

2. *Page 2*

**Explanation and modification:**

"translational landslides" in line 28 means that this type of landslide, which had been developed in the Ba river basin of the Qinba–Longnan mountain area (Fan, 2007; Xu et al., 2010), not just only one.

"sand and mudstone" in line 35 should be "sandstone and mudstone", which describe the regional geology condition of this landslide in generalized, including sandstone, mudstone and siltstone. "narrow" in line 37 should be "thin", and "rock layer inclination angle" in line 37 should be "inclination angle of rock bed".

3. *Page 3*

**Modification:**

The sentences from line 43 to 53 had been rephrased and revised by English-speaking editing, and "scholars" in line 37 should be "scholars and researchers", which is more accurate.

4. *Page 4*

**Explanation and modification:**

The sentence from line 82 to 83 means that these translational landslides may be mistaken as collapse due to its special shape, and "hence, the dangers posed by such kind of landslides were easily ignored" in this sentence may not be accurate, and we deleted it and added "instead of focusing on the hidden dangers associated with landslides".

As you said, the sentence from line 88 to 89 means is not so accurate. We revised like this, "long-term on-site monitoring data and related analysis except the remote observation based on synthetic aperture radar (SAR) or satellite for such landslides, have not been reported in publications according to literature review", we focus on the

on-site and field monitoring in this paper.

**5. *Page 5**
**Modification:**

"rear crack" in line 90 had been corrected to "crown crack".

"research" in line 94 had been corrected to "we selected".

**6. *Page 6**
**Modification:**

As you said "This landslide is common" in line 116 is confusing and ambiguity, we had corrected to "This landslide occurred".

"Forming Conditions" in line 123 should be corrected to "Formation Conditions", and we want to analyze the characteristics and reasons of formation for this landslide.

"Engineering Geology Characteristics" in line 125 can be revised as "Geometric Characteristics".

"width" in line 128 can also be revised as "length". And we added a reference for definition of landslide scale in line 130 (Ministry of Land and Resources of the PRC, 2006), and we added it in reference table.

"$\angle 6° - 8°$" in line 132 represents inclination degree of the rockbed is between 6° to 8°. If it does not meet international standards, we can write like this, "the inclination degree of the rockbed is $6° \sim 8°$".

"which is a typical nearly horizontal consequent bedding rock slope" in line 132 and 133 should be corrected to "which is a gently inclined bedding rock landslide".

"sectional graph" in line 132 had be corrected to "cross section graph",

**7. *Page 7**
**Modification:**

We are very sorry about this figure,which lacks some information due to confidentiality issue, The formation lithology information has been added in the figure, and the elevation information of the main part is also displayed in the section view.However, this figure is mainly to show our work and provide data support for subsequent numerical simulation, so we have changed the "topographic map" to "schematic map".

We are very sorry about this issue, as shown in Figure 3, the right side of the figure points to 249 °,but 249 ° points to the left of the figure 2, that's why the plan view is the opposite of the section view, in fact, these pictures are all correct, If we want to correspond exactly, as follow next figure.

[Figure]

"Fig. 3 I-I' sectional graph of the landslide" in line 141, we changed it to "Fig. 3 I-I' cross section of the landslide". As you said that it is difficult to be a translational landslide, we also regarded it as a bedrock collapse during the preliminary investigation. However, we observed that the plate-shaped main bodies is cut by the crown cracks, the houses had cracked at the front edge, and the roadbed is pushed uplifted at the front edge, which are shown in Fig. 2. Considered the Xima middle school landslide (Xu et al., 2010), Dahe middle school landslide (Li, 2009) in Nanjiang County, Bazhong City, and the Tiantai township landslide in Xuhuan County (Fan et al., 2008), the Wobaoshi landslide and above-mentioned landslides are all located in the red beds of the Qinba–Longnan mountainous area, and the characteristics and genetic mechanism are roughly the same. After consulting with Professor Xu, we defined the Wobaoshi landslide as a special type of translational landslide.

"longitudinal length is much less than the lateral width" in line 143, the description of landslide shape is not clear, we changed it to "the landslide is in a flat shape integrally, and the lengthwise is considerably smaller than the crosswise on the plane".

"multistage dangerous rock mass with deformation" in line 145 is ambiguous, we changed it to "a bedrock collapse".

"during disaster investigation" in line 145 means the investigation of geological hazard, which is funder of this manuscript, *CGS of China Geological Survey Project (Geological Disaster Investigation and Monitoring in Bazhong)*. And we changed it to "during investigation of geological hazard".

Xu, Q., Fan, X., Li, Y., and Zhang, S.: Formation condition, genetic mechanism and treatment measures of plate-shaped landslide, Chinese Journal of Rock Mechanics and Engineering, 29(2):242-250, 2010.

Li, Y.: Study on formation mechanism and prevent methods of plate girder landslide. Chengdu University of Technology, 2009.

Fan X., Xu Q., Zhang Z., Meng, D and Tang, R.: Study of genetic mechanism of translational landslide, Chinese Journal of Rock Mechanics and Engineering, 27(Supp.2):3753-3759, 2008.

8. *Page 8*

**Explanation and modification:**

"a certain degree of aperture" in line 149 means that the structural surface of sliding body had cracks, may be "a certain degree of aperture" is confused, we had corrected to "the structural surface of the sliding body contain cracks".

"and the bottom of the crack is filled with clay in addition to gravel and collapse debris" in line 150 and 151, this phenomenon was obtained through field investigations, and the first author (Yimin Liu) participated in the investigation and also according to the investigation report of Wobaoshi landslide, maybe we should added this reference here.

"obvious" in line 152 had been corrected to "main".

"Then, the plate-shaped sliding bodies I and II were formed." in line 155 had been corrected to "and the plate-shaped sliding bodies I and II are also shown in Fig. 2."

"two-stage" in line 155 means that this landslide has two plate-shaped sliding bodies.

"the pore-water in the cracks can be observed" in line 160, you want to know how we can observe the pore-water. When we did a field investigation of the landslide, we discovered water remain in the cracks during rainy season, especially during large rainfall, thus we deduced the cracks will have preferable water-storage conditions, and it can be found in the investigation report of Wobaoshi landslide (Chen et al.,2015). Then we installed pore-water pressure gauges to monitoring the pore-water level.

"trailing edge" in line 162 means crown of the landslide, and this glossy is not professional, we changed it to "crown" (Highland and Bobrowsky, 2013).

"plate-shaped landslide" in line 173, this glossary was first put forward by Professor Xu and Professor Fan (Xu et al., 2010), and I added this reference after this sentence.

Xu, Q., Fan, X., Li, Y., and Zhang, S.: Formation condition, genetic mechanism and treatment measures of ==plate-shaped landslide==, Chinese Journal of Rock Mechanics and Engineering, 29(2):242-250, 2010.

Lynn Highland, Peter Bobrowsky. The Landslide Handbook—a Guide to Understanding Landslides: A Landmark Publication for Landslide Education and Preparedness[M]. Springer Berlin Heidelberg, 2013.

9. *Page 9*

**Explanation and modification:**

"the pore-water in the cracks exists" in line 188, our explanation is the same as the line 160. Maybe here are some repetitions with the line 160, and we deleted this sentence.

"multilevel plate girders" in line 193 is inconsistent with the previous glossary in line 19, and we corrected to "multi-stage main bodies".

**10. *Page 12**
**Modification:**

The font dimension in Fig.7 was so small in previous vision, and we have enlarged the font dimension both in Fig.7(a) and Fig.7(b), then Fig.7 would be readable.

"rainfall intensity" in line 241 and 243 is wrong, and we have revised it as "amount of rainfall".

"absolute stretching amount" in line 246 and 248 is not so clear, and we have revised it as "absolute value of extension", and also revised in line 240, 248 and 331. And the absolute value of extension is shown in Table 3.

**11. *Page 13**
**Modification:**

"Absolute slippage amount curves of crack I and II" in line 252 in Fig.8 is not so correct, and we have revised it as "Curves of the absolute extension value of crack I and II".

"the Wobaoshi landslide is still moving along the sliding surface" in line 257 and 258 is not accurate here, and this conclusion which only rely on absolute extension value of crack I and II is not reliable, and this conclusion should be obtained by the model calculation and numerical simulation in chapter 3. Therefore, we deleted this sentence in line 258.

"rainfall intensity" in line 258 is wrong, and we have revised it as "amount of rainfall".

"replenishment" in line 263 means the pore-water supplement by rainfall, we should revised it as "rainfall".

"plate girders" in line 263 is inconsistent with the previous glossary in line 19 and 193, and we revised it to "main bodies". And we also revised it to "body" or "plate-shaped body" in other sentences in revised manuscript.

**12. *Page 14,15**
**Explanation and modification:**

The chapter 3 had been revised, especially the terminology and ambiguity. The sentence "that is, the problem's solution is solved by analyzing the destruction of the soil's balance" in line 284 and 285, means that stability calculation of the main bodies is based on the limit equilibrium method, and basic characteristic of this method is the Mohr-Coulomb failure criterion, and we add a reference here about the Mohr-Coulomb failure criterion in line 285.

We have done a proofreading on Page 15.

**13. *Page 16, 17, 18 and 19**
**Modification:**

"sectional graph" in line 312 have be changed to "cross section".

"trailing edge" in line 323 have been changed to "rear".

"landslide monitoring engineering" in line 327 and 328 have been changed to "long-term monitoring".

The font dimension in Figs. 10 and 11 was so small in previous vision, and we have enlarged the font dimension both in Figs. 10(a), 10(b), 11(a) and 11(b), then Figs. 10 and 11 would be readable.

The sentences from line 358 to 361 are not so clear, we have revised like this, "The left boudary and right boundary are about 30m from the body I and II respectively, and the lower boundary location is at sea level."

We have increased the font dimension in Fig. 12. And "Sliding bodies occur when pore-water level increase" in line 379 in Fig. 12(b) have been changed to "Tilt and slide occurs when pore-water level increase".

14. *Page 20*

**Modification:**

"Fig 12(b) shows that, under the combined effect of the pore-water pressure and seepage, the multistage girders slide horizontally along the sliding surface." from line 384 to 386, we have revised them as "Fig 12(b) shows that, under the combined effect of the pore-water pressure and seepage, the multistage main bodies deform horizontally along the sliding surface."

"this special structure of translational landslide widely occurs in the Qinba–Longnan mountainous area during the investigation of geological hazard hidden dangers" from line 393 to 394, we have revised them as "this special structure of translational landslide widely develops in the Qinba–Longnan mountainous area during the investigation of geological hazard".

The title "Failure Mode Exploration" in line 403 should been changed to "Failure Mode Analysis".

14. *Page 20-34*

**Explanation and modification:**

We have done a proofreading on the chapter 4 and 5, and the chapter 4 and 5 have been rephrased according to your comments and the suggestions of English-speaking editing agency (enago).

---

## Author Response (AR4)

**Reply to editor**

Dear editor,

Many thanks for your great and careful comments, and we thank you for giving us an opportunity to revise this manuscript. According to your comments and requirements, we have checked the manuscript and correct mistakes, and delete some repeats.

With help of an English-speaking expert, we focus on revise structure and language of the manuscript, to meet international standards, especially in introduction, landslide location in section 1.1, data analysis in section 2, model calculation in section 3.1, discussion in section4 and conclusions in section 5. And the Figs. 2, 11 and 12 have been remade and revised according to your comments, we merge Figs. 4 and 5 in previous version into one figure for brevity, and we merge Figs. 6 and 7 in previous version into one figure to show the relationship between the monitoring data. And we add a new figure to show the installation schematic of water pressure gauge, rain gauge and crack meter in Fig. 5. And we have changed "pore-water level" to "accumulated water level in crack", changed "crack slippage" to "Opening width of crack" in the new version.

Please see the detailed revision for point-by-point reply, and the modified parts are marked in red in the revised manuscript. We believe that the quality of the paper has been greatly improved after this revision, and we look forward to hearing from you.

Best regards,
Yimin Liu, Chenghu Wang, Guiyun Gao, and Pu Wang, et al.

**Detailed revision for editor's comments**

1. *Page 2, 3, 4*

**Explanation and modification:**

The modified parts are marked in red according to comments in line 35, 37 and 39, and we delete the reference (Guzzetti et al., 2004) in line 42.

Our study belongs to the first category in line 43, and we elaborate our viewpoint in the conclusions.

We add some references from line 54 to 56, and we reconstructed the sentences in red from line 60 to 67.

The modified parts are marked in red in line 70, 75, 84, 85 and 87.

The sentences about the study focuses on have been reconstructed and optimized from line 79 to 89. And we revise it as "we found just a few field surveys and monitoring data for this type of landslide" in line 85.

2. *Page 5, 6, 7 and 8*

**Explanation and modification:**

The paragraph about landslide location in section 1.1 have been reconstructed and optimized. We explain the Shilong river is the second grade tributary of the Ba river in line 108. We add references to explain red beds in China in line 110.

We adjust the map scale in Fig 1, and the location of the landslide that is not so small in Fig 1, meanwhile the Fig 1 presents the elevation information, geographical position, and longitude and latitude data of the landslide. Therefore, we believe that Fig 1 should be preserved after revision. We also add a geographic map in Fig 1 to show the specific location of the landslide here, whether it can be added to Fig 1 in your opinion.

[Figure]

Fig. 1 Geographic map of the Wobaoshi landslide

The paragraph about landslide characteristics in section 1.2 have been reconstructed and optimized. We simplify headline of section 1.2, and delete subchapter in line 128 in previous version. We delete the sentence in line 125 in previous version because the dynamic change of groundwater has little relation with our analysis. We add information of the remote sensing data and the reference in line

122. We add references to explain Nanyangchang anticline in Daba mountain in line 129 in Fig.2.

And the Figs. 2 has been remade and revised according to the comments, mainly enlarge the map and reduce the space of legend to meet the comments, and revise the title of Fig.2.

We simplify headline of chapter 2.

3. *Page 9, 10, 11, and 12*

**Explanation and modification:**

The paragraph about monitoring scheme in section 2.1 have been reconstructed and optimized.

We merge Fig.4 and 5 into one figure, and revise the title of Fig.4 in line 174 Fig.4 in this version shows the location and photos of the monitoring equipment.

We acquired the depth data of the cracks I and II from the investigation report of Wobaoshi landslide (Chen et al., 2015), and we measured and estimate $H$ by laser range finder in field survey, and we add the reference about the depth of the cracks in line 195 and 196.

We add a new figure to show the installation schematic of water pressure gauge, rain gauge and crack meter in Fig. 5 in line 199.

The comment in line 210 in previous version is that the width of crack I don't seem to be 5m in Fig. 2(e). The reason is that the photo of crack I in Fig. 2(e) is not the widest position, we choose a wider position of crack I to monitor its width, in additional, as shown in Fig. 5, the initial width of crack I gauge is approximately 5m, which was measured by the gauge, the installation location of the sensing system and target also enlarge distance of the measurement result.

The specific data analysis in section 2.2 has been reconstructed and optimized from line 203 to 229, and we merge Figs. 6 and 7 in previous version into one figure to show the relationship between the monitoring data.

The sentences about the significance of chapter 3 have been reconstructed and optimized from line 232 to 239.

5. *Page 13 14, 15 and 16*

**Modification:**

The sentences about the ideal elastic-plastic model have been reconstructed and optimized from line 245 to 249.

The sentences in section 3.1 have been reconstructed and optimized from line 258 to 260. In line 263, $h_{cr2}$ is the critical height of the pore-water levels in crack II when $K_2$ is set to 1 in Eq. (1). And we define $h_{cr}$ as the critical pore water level.

The sentences about the internal friction angle of the surface have been reconstructed and revised from line 280 to 282.

The sentences about calculation results about the critical water level have been reconstructed and revised from line 283 to 289. , and explanation of the $h'_{cr}$(measured)

close to $h_{cr}$ (theoretical) can be seen in section 4.2 of discussion.

In line 303, the curves of the hc1-k1 and hc2-k2 in Fig. 10 represent Eqs. (3) and (1), respectively.

**6. *Page 17 and 18**
**Explanation and modification:**

These are boundary conditions in the numerical model of left and right boundaries in line 315, and the lower boundary was located at sea level to eliminate the boundary effect in line 316.

According to the comments of Fig. 11, we have modified the legends by using the same scale, added Fig. 11 (d) to correspond to 5 steps in Table 5, added the description of in Fig. 11(d) and (e) from line 377 to 379, and added the X and Y directions in Fig. 11. And we also revise the title of Fig. 11.

**7. *Page 19, 20, 21 and 22**
**Explanation and modification:**

We add significance of the discussion from line 362 to 364.

We have corrected the mistakes and imprecisions in line 370, 372 and 374.

According to the comments of Fig. 12, we add a figure into Fig. 12 to show the sequence of movement of bodies. The sequence is that the outermost body II slides firstly, then the balance of water pressure in cracks is broken, and then this condition causes the sliding of the body I. We add the description about Figs. 12(b) and 12(c), to describe failure mode of the Wobaoshi landslide.

We have reconstructed and simplified the sentence from line 396 to 401, because this sentence was too long before.

The description about internal friction angle has been reconstructed and revised from line 404 to 406, and the sentence about the relationship between $h_{cr}^{'}$ and $h_{cr}$ also has been reconstructed from line 421 to 424.

We have revised the sentence about optimization monitoring methods from line 426 to 428.

**8. *Page 23 and 24**

We have completely reconstructed and rephrased the conclusion in chapter 5, and marked in red from line 449 to 472.

**9. *Page 32, 33 and 34**

We have revised the names of Figs. 1 and 2.

The lithology in Table 5 has been revised to consistent with the names in Fig. 3.

The comment in line 649 in previous version is that where did you get these numbers from, we think that the numbers of water level value are obtained from the

measured data in Table 3. Note these water levels are the absolute value (presented in elevation), while that in Table 3 are level value related to main bodies.